# Paternal SARS-CoV-2 infection impacts sperm small noncoding RNAs and increases anxiety in offspring in a sex-dependent manner

Given that the SARS-CoV-2 virus, and the COVID-19 pandemic, constitutes a major environmental challenge faced by billions of people worldwide, we investigated whether paternal pre-conceptual SARS-CoV-2 infection has impacts on sperm RNA content, and intergenerational (F1) and transgenerational (F2) effects on offspring phenotypes. Using an established mouse-adapted SARS-CoV-2 (P21) preclinical model, we infected adult male mice with the virus, or performed a mock control infection, and bred them with naïve female mice four weeks later, when males were no longer infectious. Here we show that offspring of infected sires display increased anxiety-like behaviors. Additionally, the F1 offspring have significant transcriptomic changes in their hippocampus. Various sperm small noncoding RNAs, including PIWI-interacting RNAs, transfer-derived RNAs and microRNAs, are differentially altered by prior paternal SARS-CoV-2 infection. Microinjection of RNA from the sperm of SARS-CoV-2 infected males into fertilized oocytes leads to a phenotype resembling that of the naturally born F1 offspring, supporting the interpretation that sperm RNAs are contributing to the outcomes of our paternal SARS-CoV-2 model. Therefore, this study provides evidence that paternal SARS-CoV-2 infection impacts sperm and affects offspring phenotypes. These findings have public-health implications and inform further research in males affected by COVID-19, and their offspring.

Psychiatric disorders, such as anxiety and depressive disorders, are amongst the leading causes of disease burden worldwide, yet their aetiologies are not completely understood[1,2]. The problem of 'missing heritability' has arisen in many studies which show that genetic factors alone do not entirely account for the inheritance of these disorders[3]. This poses serious challenges for developing effective therapeutic and preventative strategies to reduce the global burden of mental health disorders. On the other hand, there is growing evidence suggesting that paternal and maternal environmental exposures, including stress[4–6], dietary changes[7,8], and toxins[9–11], can lead to maladaptive

changes in the mental health of offspring via epigenetic inheritance. Furthermore, it is becoming increasingly recognized that sperm can transfer environmentally modifiable information, particularly in the form of small noncoding RNAs, to the oocyte at conception, which can play important roles in shaping offspring development and disease susceptibility[12–15]. Many studies have now revealed that paternal pre-conceptual exposure to stress[16–18], dietary perturbations[7,13], and drugs of abuse[19,20] can alter the sperm small RNA payload and subsequently lead to changes in offspring brain and behavioral phenotypes. Furthermore, recent studies characterizing the phenotypes of offspring

✉ e-mail: carolina.gubert@florey.edu.au; anthony.hannan@florey.edu.au

after microinjection of differentially expressed sperm RNAs into fertilized oocytes demonstrate that sperm-derived RNAs are mechanistically linked to offspring fitness[18,21].

The harmful multigenerational effects of maternal exposure to viral infection and immune activation during gestation have been relatively well-explored[22–25]. Recent studies show that maternal Severe Acute Respiratory Syndrome Coronavirus 2 (SARS-CoV-2) infection during pregnancy may also affect the neurodevelopment of the offspring[26–28], although more studies are needed to confirm and extend these initial findings. However, the effects of paternal exposure to SARS-CoV-2 infection and immune activation, whilst the subject of recent speculation[29], have not been previously investigated. On this score, we previously discovered that paternal pre-conceptual *Toxoplasmosis gondii* infection in mice can modify both offspring (F1) and grand-offspring (F2) brain development and behavior via changes to the sperm noncoding RNA profiles[21]. Furthermore, we recently also revealed striking differences in the depressive behavior of offspring from sires pre-conceptually exposed to a viral-like immune challenge (induced by poly-inosinic:polycytidylic acid (Poly I:C))[30]. Additionally, we have also shown that bacterial-like immune activation in male mice produces various affective and cognitive phenotypes and altered immune response in their offspring, including reduced anxiety in the F1 female offspring[31]. These recent studies led us to question whether paternal exposure to other pathogenic infections, including viruses, can affect offspring brain and behavioral phenotypes via modifications in the sperm RNA payload. These are particularly important questions given that over 778 million people have documented infection with SARS-CoV-2 since the start of the COVID-19 pandemic (World Health Organization, June 2024), although the actual number of global cases is likely to be much higher.

Although infection with SARS-CoV-2, a positive-sense single-stranded RNA virus, often results in a mild respiratory illness in humans, approximately 15% of individuals experience a more severe disease[32]. Since there is emerging evidence that other infections can reprogram offspring mental health[21,30,31], it is imperative to consider whether infection with SARS-CoV-2 itself can affect the mental health of future generations. This could have major public health implications since many of those who have been infected with SARS-CoV-2 will have children post-infection.

We have previously found that paternal exposure to Poly I:C, a viral mimetic which causes a surge in pro-inflammatory cytokines, alters sperm noncoding RNA profiles and offspring behavioral phenotypes[30]. As such, we hypothesized that paternal SARS-CoV-2 infection will lead to changes in offspring brain and behavioral phenotypes via changes to the sperm RNA content. To address this hypothesis, we used a C57BL/6J adult male mouse model infected with P21 SARS-CoV-2, which models severe acute infection[33]. This model is advantageous because it closely mimics a moderate-to-severe human disease with the presence of lung inflammation, cytokine storm at the peak of infection (day 3 post-infection), and age-dependent disease severity[33]. Here we show that the first generation of offspring from sires pre-conceptually infected with SARS-CoV-2 (P21) display increased anxiety-like behavior. We also detected multiple changes in the small noncoding RNA content in the sperm from SARS-CoV-2 infected sires. Furthermore, microinjecting isolated sperm RNA from SARS-CoV-2 infected sires into fertilized oocytes resulted in a partial phenocopying of the anxiety-like behavior seen in the naturally born F1 offspring. We demonstrate that SARS-CoV-2 infection alters the molecular composition of sperm, and the offspring phenotype, and has important intergenerational implications for human health, including predisposition to relevant brain disorders.

## Results

### SARS-CoV-2 infection in male mice results in bodyweight loss without significantly affecting reproductive and litter characteristics

Male C57BL/6J mice were infected at 8 weeks of age via intranasal administration of SARS-CoV-2 P21 virus ($10^4$ TCID50 (median tissue culture infectious dose) per mouse in 30 µl). Mock infection of the control group was carried out with PBS. A significant infection x time interaction showed that SARS-CoV-2 infected mice experienced a reduction in bodyweight ($F_{(10, 179)} = 28.37$, $P < 0.0001$, Supplementary Fig. 1A) that was recovered by day 8 post-infection. *Post-hoc* analysis revealed that bodyweight was significantly decreased from day 2 ($P = 0.0059$) to day 7 ($P = 0.0048$) post-infection. This response replicates the previous findings of this SARS-CoV-2 P21 infection model as weight loss is a strong indication of infection[33]. At 4 weeks post-infection, male mice that were either previously infected with SARS-CoV-2 (and lost at least 9% of their bodyweight at day 3 post-infection) or were mock infected (controls), were mated with naïve female mice. It should be noted that at this 4-week timepoint all mice had already cleared the infection for several weeks[33], and thus the naïve female mice were never exposed to the virus.

At 4 weeks post-infection, the testes weight of the SARS-CoV-2 infected mice was comparable to mock infected mice ($U = 95$, $P = 0.333$, Supplementary Fig. 1B). Additionally, litter sizes ($F_{(1, 17)} = 0.025$, $P = 0.877$, Supplementary Fig. 1C), the number of viable litters produced, and litter sex ratios ($F_{(1, 17)} = 0.035$, $P = 0.854$, Supplementary Fig. 1D) were comparable between SARS-CoV-2 infected and control male mice. Furthermore, histological examination of the testes revealed no obvious differences in cytology between control and SARS-CoV-2 infected male mice at 4-weeks post infection (Supplementary Figs. 2A, B). There were also no differences in average numbers of spermatocytes ($U = 11$, $P = 0.841$, Supplementary Fig. 2C), round spermatids ($U = 7.5$, $P = 0.341$, Supplementary Fig. 2D), and elongated spermatids ($U = 8$, $P = 0.421$, Supplementary Fig. 2E) in stage III seminiferous tubules from control and SARS-CoV-2 infected mice. Finally, we investigated whether the maternal fecal microbiota had changed after exposure to either mock or SARS-CoV-2 infected males (previously infected) during the mating period, as this is known to affect offspring outcomes[34]. There were no differences observed in α-diversity (Shannon Index) between female mice bred with SARS-CoV-2 infected and mock infected male mice ($t = 0.369$, $P = 0.719$, Supplementary Fig. 2F).

### Paternal SARS-CoV-2 infection affects F1 offspring bodyweight trajectories

Although it initially appeared that the male and female offspring of infected sires had increased bodyweight at postnatal day (PND) 8, this trend did not reach significance ($F_{(1, 11.97)} = 4.655$, $P = 0.052$, Supplementary Fig. 1E) after adjusting for litter effects. There were no significant paternal treatment effects for offspring bodyweight at PND 15 ($F_{(1, 12.08)} = 3.128$, $P = 0.102$, Supplementary Fig. 1F) and PND 22 ($F_{(1, 12.01)} = 3.991$, $P = 0.069$, Supplementary Fig. 1G) after adjusting for litter effects. However, there was a significant paternal treatment x time x sex interaction for the bodyweight trajectories of F1 mice after weaning ($F_{(8, 568)} = 5.802$, $P < 0.0001$, Supplementary Fig. 1H). *Post-hoc* analysis revealed that offspring of infected sires were significantly heavier than offspring of control sires at 4 weeks of age ($P = 0.014$).

### Paternal SARS-CoV-2 infection increases F1 offspring anxiety-like behaviors

A range of behavioral tests related to anxiety, depression, locomotion, learning, memory and sociability were performed in the F1 offspring from 8 weeks of age as shown in Fig. 1A. Behavioral z-scores were also calculated for each behavioral parameter and averaged within each

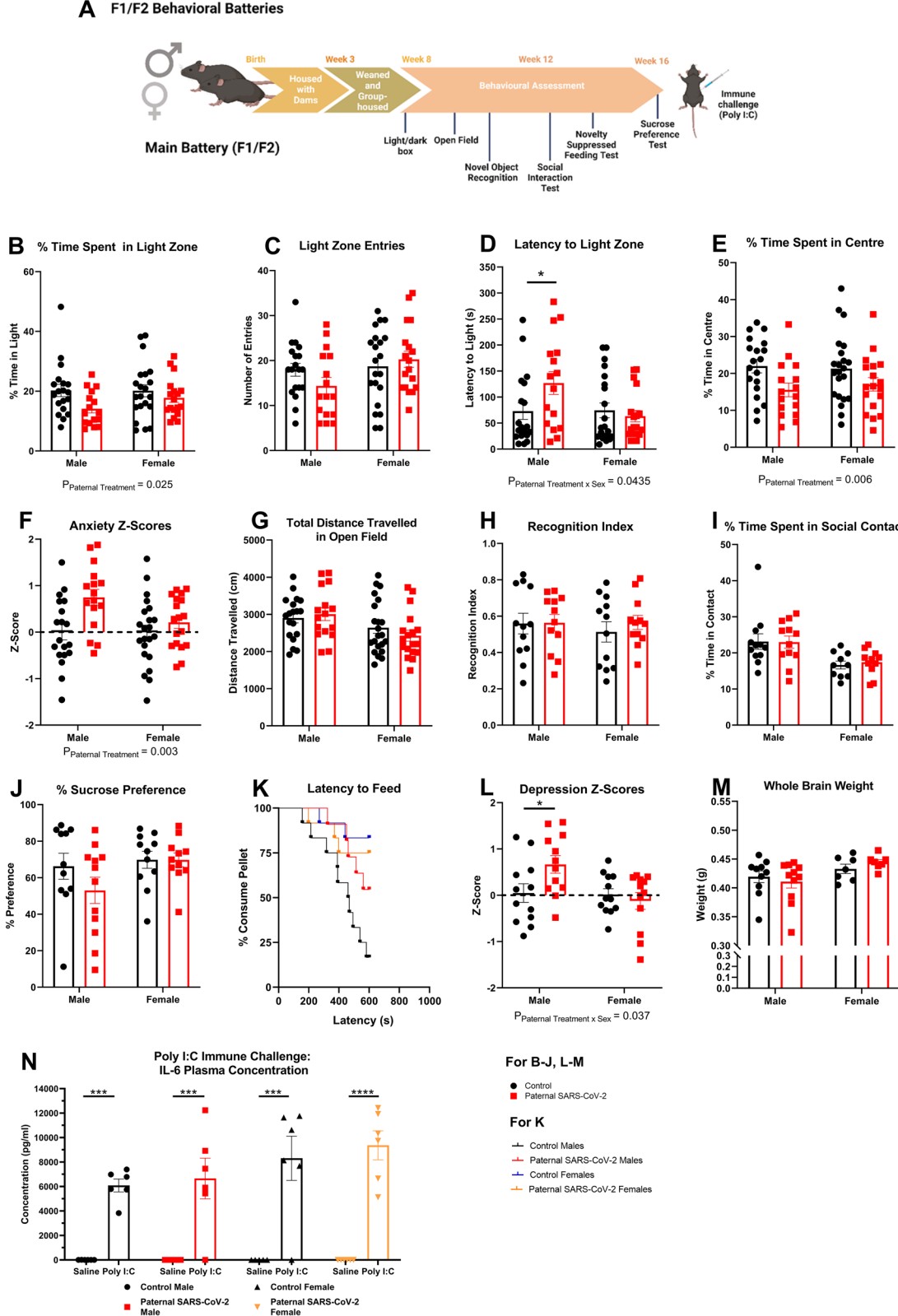

behavioral domain. In the anxiety tests, offspring of infected sires showed significantly decreased time spent (%) in the light zone of the light-dark box ($F_{(1, 70)} = 5.236$, $P = 0.025$, Fig. 1B), but no significant changes in light zone entries ($F_{(1, 70)} = 0.338$, $P = 0.563$, Fig. 1C). There was also a significant paternal treatment x sex interaction for the latency to enter the light zone ($F_{(1, 70)} = 4.227$, $P = 0.0435$, Fig. 1D). Post-hoc analysis revealed that only the male offspring of infected sires had

a longer latency time to enter the light zone of the light-dark box ($P = 0.022$). Paternal SARS-CoV-2 offspring also spent significantly less time (%) in the centre zone of the open-field test ($F_{(1, 71)} = 7.876$, $P = 0.006$, Fig. 1E). The behavioral z-scoring for the anxiety domain revealed a significant main effect of paternal treatment such that the paternal SARS-CoV-2 offspring had significantly increased anxiety scores ($F_{(1, 71)} = 9.166$, $P = 0.003$, Fig. 1F). Furthermore, there was no

**Fig. 1 | Paternal SARS-CoV-2 infection significantly changes anxiety-like behavior in F1 offspring. A** Behavioral battery and timeline for the main F1 and F2 cohorts. Paternal SARS-CoV-2 infection significantly decreases (**B**) % time spent in the light zone of the light-dark box ($n = 19$ CON M, $n = 21$ CON F, $n = 16$ P.SARS M, $n = 18$ P.SARS F) without changing the (**C**) number of entries into the light zone ($n = 19$ CON M, $n = 21$ CON F, $n = 16$ P.SARS M, $n = 18$ P.SARS F) for F1 offspring. Paternal SARS-CoV-2 infection significantly increases (**D**) the latency to enter the light zone in the male offspring only ($P = 0.022$, general linear model with Bonferroni correction)($n = 19$ CON M, $n = 21$ CON F, $n = 16$ P.SARS M, $n = 18$ P.SARS F) and significantly decreases the (**E**) % time spent in the centre of the open-field ($n = 19$ CON M, $n = 22$ CON F, $n = 16$ P.SARS M, $n = 18$ P.SARS F) while increasing the (**F**) overall anxiety behavioral z-scores ($n = 19$ CON M, $n = 22$ CON F, $n = 16$ P.SARS M, $n = 18$ P.SARS F) for F1 offspring. Paternal SARS-CoV-2 has no significant effects on (**G**) the total distance traveled in the open-field ($n = 19$ CON M, $n = 22$ CON F, $n = 16$ P.SARS M, $n = 18$ P.SARS F), (**H**) the recognition index in trial 2 of the novel-object recognition test ($n = 12$), (**I**) % time spent in contact with the guest mice over 10 min ($n = 12$ CON M, $n = 10$ CON F, $n = 12$ P.SARS M, $n = 11$ P.SARS F), (**J**) % pre-ference for sucrose in the sucrose preference test ($n = 11$ CON M, $n = 11$ CON F, $n = 12$ P.SARS M, $n = 11$ P.SARS F), and (**K**) latency to feed in the novelty-suppressed feeding test for F1 offspring ($n = 12$ CON M, $n = 12$ CON F, $n = 11$ P.SARS M, $n = 12$ P.SARS F). Paternal SARS-CoV-2 significantly alters (**L**) overall depression behavioral z-scores ($P = 0.023$, general linear model with Bonferroni correction)($n = 12$), while not affecting (**M**) whole brain weight ($n = 10$ CON M, $n = 7$ CON F, $n = 11$ P.SARS M, $n = 7$ P.SARS F), and (**N**) F1 plasma IL-6 levels at 2-h post-Poly I:C injection (12 mg/kg) ($n = 6$ CON M saline, $n = 6$ CON M poly I:C, $n = 5$ CON F saline, $n = 6$ CON F poly I:C, $n = 6$ P.SARS M saline, $n = 6$ P.SARS M poly I:C, $n = 6$ P.SARS F saline, $n = 6$ P.SARS F poly I:C)($P$ value from left to right: $P = 0.0004$, $P = 0.0002$, $P = 0.0004$, $P < 0.0001$, general linear model with Bonferroni correction). Data presented as mean ± SEM. General linear models and linear mixed models were used with *post-hoc* analyses where appropriate (Bonferroni-Holm corrected) except for **K**. Cox regression with proportional hazards was used to analyse **K**. Each n number refers to the number of individual animals per group. $P_{\text{Paternal Treatment}}$ = main effect of paternal treatment, $P_{\text{Paternal Treatment x sex}}$ = interaction effect of paternal treatment by sex. *$P < 0.05$, ***$P < 0.001$, ****$P < 0.0001$. Created in BioRender. Kleeman, L. (https://BioRender.com/gaza9wr).

significant change in the total distance traveled in the open-field ($F_{(1, 15.85)} = 0.025$, $P = 0.877$, Fig. 1G). Collectively these data indicate that offspring from sires previously infected with SARS-CoV-2 display significantly increased anxiety-like behavior. Furthermore, time spent (%) in the light zone, light zone entries, and the latency to enter the light zone, all indicated a robust anxiety-like behavioral phenotype was also present in a separate F1 replication cohort (Supplementary Fig. 3A–C).

Additionally, we assessed memory and sociability in the F1 offspring by performing the novel object recognition test and social interaction test respectively. In terms of memory, there were no significant differences in novel object recognition index ($F_{(1, 44)} = 0.318$, $P = 0.576$, Fig. 1H) nor were there changes in the cognition z-scores ($F_{(1, 44)} = 0.322$, $P = 0.573$, Supplementary Fig. 1I), for the F1 offspring. There were also no changes detected in sociability as the offspring from infected and control sires spent a similar amount of time in contact with the guest mice ($F_{(1, 41)} = 0.035$, $P = 0.853$, Fig. 1I), percentage time distant from the guest mouse ($F_{(1, 41)} = 0.072$, $P = 0.789$, Supplementary Fig. 1J), and made a similar number of approaches towards the guest mouse ($F_{(1, 41)} = 0.891$, $P = 0.350$, Fig. S1K). Sociability z-scores were also not significantly different between the offspring of infected and mock infected sires ($F_{(1, 41)} = 0.552$, $P = 0.462$, Supplementary Fig. 1L).

In terms of depression-like behavior, we saw no differences in percentage sucrose preference for the F1 offspring ($F_{(1, 10.53)} = 0.589$, $P = 0.46$, Fig. 1J), indicating that there are no differences in anhedonia. However, there was a trend towards decreased food consumption over 8 h, after the 18-h fast associated with this test ($F_{(1, 42)} = 4.02$, $P = 0.051$, Supplementary Fig. 1M).

The novelty-suppressed feeding test assesses depression-like behavior by measuring how much time it takes a food-deprived mouse to initiate feeding in a novel, anxiogenic arena[35]. There were no significant differences observed in the F1 offspring's latency times to feed on the chow pellet in the novelty-suppressed feeding test (treatment-hazard ratio = 0.546, $P = 0.188$, Fig. 1K). Despite no significant differences in the individual assays measuring depression-like behavior, the behavioral z-score revealed a significant paternal treatment x sex interaction for depression-like behavior overall ($F_{(1, 40.19)} = 4.598$, $P = 0.038$, Fig. 1L). *Post-hoc* analysis showed that only male offspring of infected sires had increased depression-like behavioral z-scores ($P = 0.023$). However, this subtle depression-like behavior was not seen in the separate F1 replication cohort (Supplementary Fig. 3H–J). Interestingly, there was a paternal treatment x sex interaction for the percentage change in bodyweight after the 24-h fasting period associated with this test ($F_{(1, 44)} = 0.001$, $P = 0.001$, Supplementary Fig. 1N). *Post-hoc* analysis revealed that the male offspring of infected sires lost

a smaller percentage of their original bodyweight after the 24-h fast ($P < 0.0001$). This was not accompanied by significant changes in individual food consumption over 5 min following the fast ($F_{(1, 10.53)} = 2.281$, $P = 0.161$, Supplementary Fig. 1O).

Additionally, there were also no significant differences in whole brain weight for the F1 offspring ($F_{(1, 8.7)} = 0.021$, $P = 0.888$, Fig. 1M). In terms of their pro-inflammatory immune response, neither the F1 male offspring ($F_{(1, 20)} = 0.106$, $P = 0.749$, Fig. 1N) nor the F1 female offspring ($F_{(1, 19)} = 0.217$, $P = 0.646$, Fig. 1N) showed differences in their plasma IL-6 levels at 2 h after a Poly I:C (viral mimic) immune challenge (12 mg/kg).

## Paternal SARS-CoV-2 infection significantly modifies sperm small noncoding RNA profiles

There is growing evidence that sperm small noncoding RNAs may play a role in reprogramming offspring phenotypes on the paternal side[36]. Indeed, various environmental challenges, such as stress and parasitic infection, have been shown to alter sperm small noncoding RNA profiles and these changes have been directly linked to offspring phenotypes[18,21]. Since we saw a change in offspring anxiety-like behavior due to paternal SARS-CoV-2 infection, we investigated whether there were any associated changes in the sperm small noncoding RNA of the infected male mice at the time of conception (4 weeks post-infection).

After correcting for false discovery (FDR < 0.05), we found 4 significantly downregulated piRNA clusters identified using the cluster prediction tool proTRAC in sperm[37]. These included cluster 272 ($P = 0.000048$), cluster 1675 ($P = 0.000048$), cluster 75 ($P = 0.000048$), and cluster 678 ($P = 0.000048$)(Fig. 2A–C). Interestingly, all of these differentially expressed piRNA clusters aligned with the Clusterin (Clu) gene on chromosome 14 (Fig. 2A–C). A list of the top 50 clusters and their corresponding genomic coordinates can be found in Supplementary Table 1. In addition, we found two significantly upregulated small noncoding RNAs in the sperm, including miR-3471 ($P = 0.034$) and pro-TGG-3-1 ($P = 1.14 \times 10^{-11}$) (Fig. 2D).

To look at predicted gene targets of differentially expressed miRNAs, we applied miR-3471 and pro-TGG-3-1 to miRDB and TargetScan (TargetScan Mouse Custom Release 5.2) databases respectively. To obtain potentially relevant gene targets, we restricted gene targets for miR-3471 to only those with a target score above 90 and that are known to be expressed in the early stages of embryonic development (e.g., zygote up to morula stages), since there is growing evidence that sperm RNAs affect early embryonic development and gene expression[36,38,39]. For miR-3471, seven predicted gene targets met these criteria whereas for pro-TGG-3-1, 77 predicted gene targets met these criteria. The 84 predicted gene targets were used in a gene

**Fig. 2 | SARS-CoV-2 infection modifies expression of small noncoding RNAs found in the mature spermatozoa at 4-weeks post-infection. A** Circular heatmap (Log$_2$ fold-change) of protract-identified total piRNA clusters (**B**) Heatmap displaying the relative expression of the differentially expressed piRNA clusters across all biological replicates (each biological replicate contains RNA pooled from 3 animals; $n$ = 5 CON, $n$ = 4 SARS). **C** MA (Bland-Altman) plot of the Log$_2$(average expression + 1) for each piRNA cluster vs. Log$_2$ fold-change relative to control values for each cluster (MA plot shows differences in expression between control and SARS-CoV-2 treatment groups for each gene) (**D**) Heatmap displaying differentially expressed miRNA and tsRNA across all biological replicates (FDR < 0.05) (**E**) Gene ontology analysis of predicted gene targets (miRNA and tsRNA) reveals pathways associated with molecular function, cellular components, and biological processes to be significantly enriched (FDR < 0.05).

enrichment analysis in Enrichr[40]. Only one KEGG pathway was significantly enriched by these targets (FDR < 0.05) which was "signaling pathways regulating pluripotency of stem cells". Ten pathways were found to be significantly enriched in the gene ontology analysis (FDR < 0.05)(Fig. 2E), some of which included "negative regulation of DNA-templated transcription", "negative regulation of nucleic acid-templated transcription", and "regulation of DNA-templated transcription".

### Grand-paternal SARS-CoV-2 infection significantly affects F2 litter sizes and F2 offspring bodyweight prior to weaning

To investigate whether SARS-CoV-2 infection in males leads to transgenerational changes in behavioral phenotypes, we bred naïve F1 male offspring from both SARS-CoV-2 and control sires with naïve females at 10 weeks of age. It is important to note that these F1 male offspring were not exposed to either behavioral testing or a SARS-CoV-2 infection. Notably, the F2 litters sired by paternal SARS-CoV-2 male offspring had significantly smaller numbers of pups ($U = 5.5$, P = 0.0315, Supplementary Fig. 4A) despite the number of viable F2 litters being comparable between the two groups. There were also no significant differences in F2 litter sex ratios ($U = 12$, P = 0.215, Supplementary Fig. 4B).

After the litter effect correction, there was a significant grand-paternal treatment x sex interaction for F2 offspring bodyweight at PND 8 ($F_{(1, 78.18)} = 7.126$, P = 0.009, Supplementary Fig. 4C). *Post-hoc* analysis revealed that only F2 male pups of infected grandsires were significantly heavier at PND 8 (P = 0.004). Although initially there was a main effect of grand-paternal treatment for bodyweight at PND 15, this effect did not survive the correction for litter effects ($F_{(1, 11.07)} = 1.127$, P = 0.311, Supplementary Fig. 4D). Nevertheless, there was a significant grand-paternal treatment x sex interaction for F2 offspring PND 22 bodyweight ($F_{(1, 73.55)} = 5.1$, P = 0.027, Supplementary Fig. 4E). However, the *post-hoc* analysis did not reveal any significant differences after the litter effect correction.

Unlike F1 offspring, there was no significant effect of grand-paternal SARS-CoV-2 infection on the bodyweight trajectories of the F2 offspring after weaning ($F_{(1, 9.8)} = 0.675$, P = 0.431, Supplementary Fig. 4F).

### Limited differences in the behavior of grand-paternal SARS-CoV-2 offspring

In the F2 (grand-paternal) offspring, a similar battery of behavioral assessments to the F1 offspring was performed. In contrast to F1 offspring, we saw no significant differences in F2 offspring light-dark box performance, including time spent (%) in the light zone ($F_{(1, 44)} = 0.127$, P = 0.724, Fig. 3A), number of entries into the light zone ($F_{(1, 9.9)} = 0.132$, P = 0.724, Fig. 3B), and the latency to the light zone ($F_{(1, 9.93)} = 1.61$, P = 0.233, Fig. 3C). Furthermore, there were no significant changes in the time spent (%) in the centre zone of the open-field ($F_{(1, 9.9)} = 0.384$, P = 0.55, Fig. 3D) or the anxiety z-score for the F2 mice ($F_{(1, 44)} = 0.863$, P = 0.358, Fig. 3E). There were also no changes in the distance traveled in the open-field for the F2 offspring ($F_{(1, 44)} = 0.099$, P = 0.754, Fig. 3F). Overall, no significant changes were seen in F2 anxiety-like behavior or locomotion.

For the novel-object recognition test, we saw no significant changes in the novel-object recognition index ($F_{(1, 44)} = 1.672$, *P* = 0.203, Fig. 3G) nor were there changes in the cognition z-score for these mice ($F_{(1, 44)} = 1.597$, P = 0.213, Supplementary Fig. 4G), indicating no differences in object memory for the F2 offspring. In terms of F2 depression-like behavior, no differences were detected in percentage sucrose preference in the sucrose preference test ($F_{(1, 9.84)} = 0.326$, *P* = 0.581, Fig. 3H). Additionally, no changes in the F2 offspring were detected for the food they consumed over 8-h after the 18-h fast associated with this test ($F_{(1, 44)} = 0.822$, *P* = 0.370, Supplementary Fig. 4H). Furthermore, no changes in the latency time to reach the

chow pellet in the novelty-suppressed feeding test were seen (treatment-hazard ratio=0.604, *P* = 0.107, Fig. 3I). Interestingly, there was a significant grand-paternal treatment x sex interaction for the depression z-score in the F2 grand-offspring ($F_{(1, 34.28)} = 6.981$, *P* = 0.012, Fig. 3J), with the *post-hoc* test revealing that the male grand-offspring of SARS-CoV-2 infected sires have a tendency towards increased depression-like behavior (*P* = 0.056). Unlike F1, no differences were seen in the percentage change in bodyweight after a 24-h fast ($F_{(1, 9.97)} = 0.01$, *P* = 0.921, Supplementary Fig. 4I) nor were there differences in the food consumed following this test ($F_{(1, 44)} = 0.136$, P = 0.714, Supplementary Fig. 4J). There were also no changes in the F2 feeding behavior z-score ($F_{(1, 44)} = 0.144$, *P* = 0.706, Supplementary Fig. 4K).Whole brain weight was also not significantly altered in the F2 offspring ($F_{(1, 18)} = 0.0003$, *P* = 0.955, Fig. 3K). In terms of the pro-inflammatory immune response, grand-paternal SARS-CoV-2 infection had no effect on F2 male offspring ($F_{(1, 20)} = 0.022$, *P* = 0.884, Fig. 3L) or F2 female offspring ($F_{(1, 20)} = 0.123$, *P* = 0.729, Fig. 3L) systemic IL-6 levels at 2-h after a Poly I:C immune challenge (12 mg/kg). Overall, no major differences in F2 behavior or immune responsivity were identified, suggesting that the transgenerational effects of SARS-CoV-2 infection may be limited.

### Microinjection of sperm RNAs from SARS-CoV-2 infected sires into fertilized oocytes alters anxiety-like behavior

To identify any possible links between the sperm RNAs of SARS-CoV-2 infected sires and the intergenerational phenotypes we have observed, we microinjected sperm RNA isolated from SARS-CoV-2 and mock-infected (control) sires into wildtype fertilized mouse oocytes. First, we used the total RNA fractions isolated from SARS-CoV-2 and control sire sperm from our previous sequencing analysis (pooled into one sample per treatment group) and performed small RNA enrichment on each sample to minimize the presence of longer RNAs <200 nucleotides in length (see Supplementary Fig. 5A for RNA fragment size distribution). These small RNA enriched samples were then microinjected into fertilized oocytes obtained from wildtype super-ovulated C57BL/6J female mice. Microinjected zygotes that survived to the 2-cell embryo stage were implanted into naïve recipient females. The resultant offspring underwent behavioral assessments similar to those in the F1/F2 main battery with the addition of tests including the elevated-plus maze (anxiety) and Y-maze (cognition), which could not be performed in the PC3 animal facility due to size constraints of the workspace in the PC3 facility.

Unlike the naturally conceived F1 offspring, the bodyweight trajectory of microinjected-SARS mice was not significantly altered from 4 weeks of age compared to microinjected-mock control mice ($F_{(1, 74)} = 0.622$, *P* = 0.433, Supplementary Fig. 5B). There were also no changes in the whole brain weight of these mice at 12 weeks of age ($F_{(1, 67)} = 1.106$, *P* = 0.297, Supplementary Fig. 5C). For the light-dark box, although there were no significant differences in the time spent (%) in the light zone ($F_{(1, 73)} = 0.019$, *P* = 0.890, Fig. 4A) or the number of entries into the light zone ($F_{(1, 73)} = 0.720$, *P* = 0.399, Fig. 4B), there was a significant microinjection x sex interaction for the latency to enter the light zone ($F_{(1, 56)} = 5.153$, *P* = 0.027, Fig. 4C). *Post-hoc* analysis revealed that only the microinjected-SARS male mice had a longer latency time to enter the light zone of the light-dark box (*P* = 0.001). Furthermore, there were significant microinjection x sex interactions for both time spent (%) in the open arms ($F_{(1, 74)} = 4.669$, *P* = 0.034, Fig. 4D) and number of entries made into the open arms ($F_{(1, 74)} = 5.333$, *P* = 0.024, Fig. 4E) of the elevated-plus maze. *Post-hoc* analyses did not reveal any significant differences for either of these parameters; however, there was a trend towards an increased number of open-arm entries for the microinjected-SARS females (*P* = 0.057). Additionally, there were no significant differences detected for time spent (%) in the centre of the open-field ($F_{(1, 73)} = 0.663$, *P* = 0.418, Fig. 4F) or distance traveled in the open-field ($F_{(1, 73)} = 0.334$, *P* = 0.565, Fig. 4H). There was

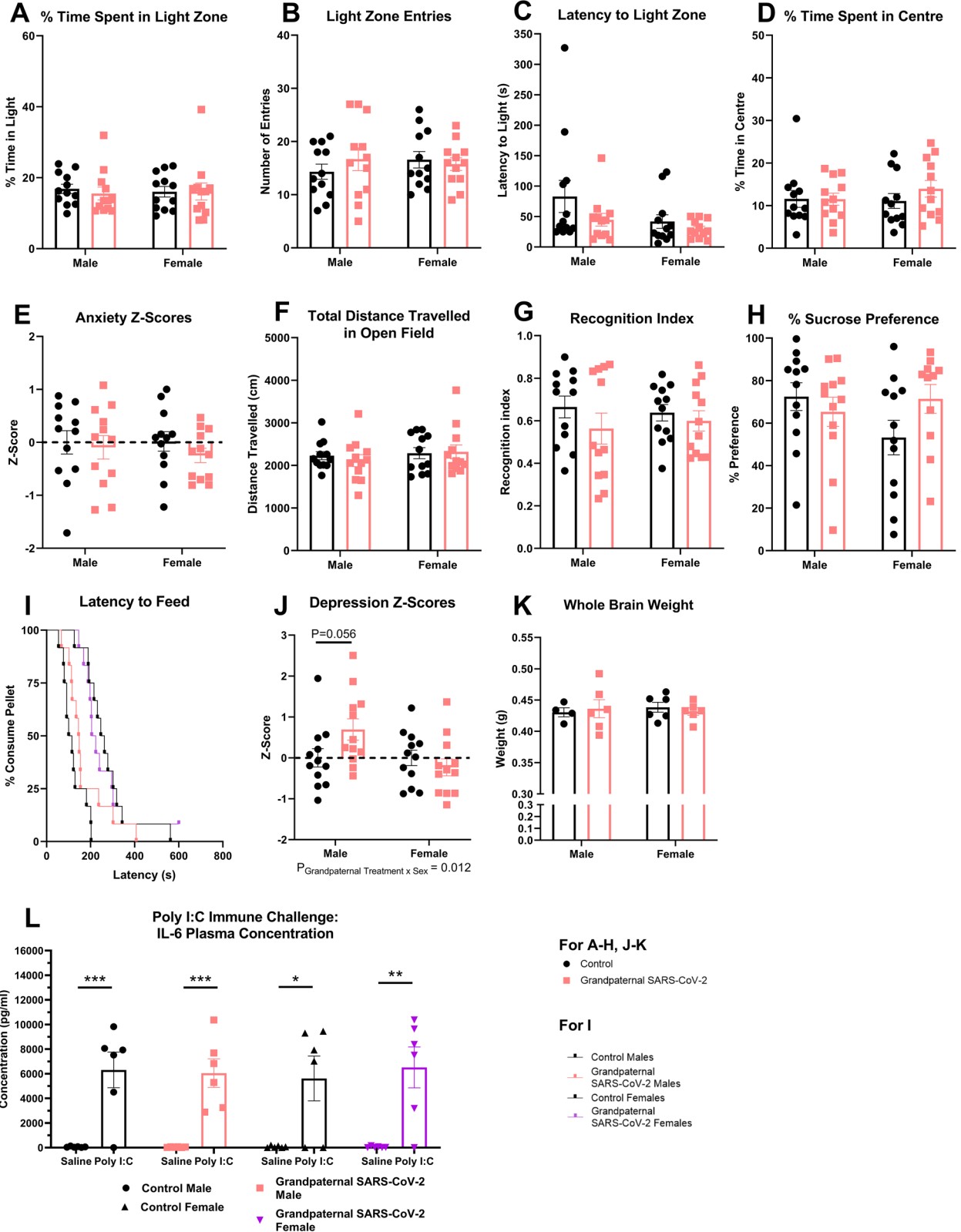

For A-H, J-K
● Control
■ Grandpaternal SARS-CoV-2

For I
Control Males
Grandpaternal SARS-CoV-2 Males
Control Females
Grandpaternal SARS-CoV-2 Females

a significantly microinjection treatment x sex interaction for anxiety z-scores ($F_{(1, 74)} = 4.999$, $P = 0.03$, Fig. 4G), with the *post-hoc* analysis revealing that microinjected-SARS males have higher anxiety levels than microinjected-SARS females ($P = 0.0132$).

In terms of cognition, there were no significant differences in the novel-object recognition index for the novel-object recognition test ($F_{(1, 72)} = 0.059$, $P = 0.809$, Fig. 4I), nor were there significant changes in

the novel-arm preference in the Y-maze ($F_{(1, 73)} = 0.639$, $P = 0.427$, Fig. S4C) or the latency to leave the home arm of the Y-maze ($F_{(1, 73)} = 0.826$, $P = 0.366$, Fig. S4D). There were no differences in the cognition z-score overall ($F_{(1, 74)} = 0.151$, $P = 0.699$, Fig. 4J). Furthermore, there were no changes in anhedonia and depression-like behavior as measured by percentage saccharin preference in the saccharin-preference test ($F_{(1, 74)} = 0.012$, $P = 0.913$, Fig. 4K) and latency time to

**Fig. 3 | Grand-paternal SARS-CoV-2 infection has little impact on the behavioral parameters measured.** Grand-paternal SARS-CoV-2 has no impact on (**A**) % time spent in the light zone of the light-dark box ($n = 12$), (**B**) number of entries into the light zone ($n = 12$), (**C**) the latency to enter the light zone ($n = 12$), (**D**) % time spent in the centre of the open-field ($n = 12$), (**E**) overall anxiety behavioral z-scores ($n = 12$), (**F**) the total distance traveled in the open-field ($n = 12$), (**G**) the recognition index in trial 2 of the novel-object recognition test ($n = 12$), (**H**) % preference for sucrose in the sucrose-preference test ($n = 12$ CON M, $n = 12$ CON F, $n = 12$ GP.SARS M, $n = 11$ GP.SARS F), and (**I**) latency to feed in the novelty-suppressed feeding test ($n = 12$). Grand-paternal SARS-CoV-2 significantly affects (**J**) depression behavioral z-scores

($n = 12$), without affecting (**K**) whole brain weight ($n = 4$ CON M, $n = 6$ CON F, $n = 6$ GP.SARS M, $n = 6$ GP.SARS F) for the F2 grand-offspring. Grand-paternal SARS-CoV-2 did not significantly alter (**L**) F2 plasma IL-6 levels at 2-h post-Poly I:C injection (12 mg/kg)($n = 6$)($P$ values from left to right: $P = 0.005$, $P = 0.0048$, $P = 0.0221$, $P = 0.0068$, general linear model with Bonferroni correction). Data presented as mean ± SEM. General linear models and linear mixed models were used with *post-hoc* analyses where appropriate (Bonferroni-Holm corrected) except for (**I**). Cox regression with proportional hazards was used to analyse (**I**). Each n number refers to the number of individual animals per group. $P_{Grandpaternal\ Treatment\ x\ sex}$ = interaction effect of grand-paternal treatment by sex.*$P < 0.05$, **$P < 0.01$, ***$P < 0.001$.

reach the chow pellet in the novelty-suppressed feeding test (treatment-hazard ratio = 1.541, $P = 0.224$ Fig. 4L) respectively. There were also no changes in the depression z-score in these mice ($F_{(1, 74)} = 0.869$, $P = 0.354$, Fig. 4M). Despite no changes in the percentage change in bodyweight after a 24-h fast ($F_{(1, 74)} = 1.284$, $P = 0.261$, Fig. 4N), there was a trend towards a microinjection x sex interaction for food consumed after a 24-h fast ($F_{(1, 74)} = 3.388$, $P = 0.070$, Fig. 4O). However, there were no changes in the feeding behavior z-scores ($F_{(1, 74)} = 0.008$, $P = 0.931$, Fig. 4P).

Overall, these data highlight that microinjection of sperm RNAs from SARS-CoV-2 infected sires into naïve oocytes may be functional in reproducing some aspects of the F1 anxiety-like phenotype. Specifically, the male mice take longer to enter the light zone of the light-dark box. Furthermore, these microinjected sperm RNAs had an overall impact on anxiety-like behavior in a sex-dependent manner in the elevated-plus maze.

## The adult offspring hippocampus transcriptome is changed by paternal SARS-CoV-2 treatment

The hippocampus is widely documented to be involved in anxiety and other affective behaviors[41,42]. Furthermore, we have previously seen changes in the hippocampus transcriptome of the offspring sired by male mice pre-conceptually exposed to a viral-like immune activation event[30]. For these reasons, we hypothesized that there would be changes in the hippocampal gene expression profiles of the F1 mice sired from male mice pre-conceptually infected with SARS-CoV-2. We used Illumina next generation mRNA sequencing of whole hippocampus samples collected from 10-week-old F1 mice (naïve to behavioral testing) to address this question. Due to the clustering of samples according to sex in the multidimensional scaling plot (MDS) analysis (Supplementary Fig. 6A), we performed the differential expression analyses separately for the F1 males and the F1 females.

For the F1 females, paternal SARS-CoV-2 infection significantly altered the expression of 20 genes in the hippocampus relative to paternal control mice (FDR < 0.05, Fig. 5A, B). One of these differentially expressed genes was upregulated and 19 were downregulated. The significantly upregulated gene was calpain 11 (*Capn11*, $P = 0.000034$). Significantly downregulated genes included prolactin (*Prl*, $P = 0.00085$), orthodenticle homeobox 2 (*Otx2*, $P = 0.000034$), collagen type VIII alpha 2 chain (*Col8a2*, $P = 0.000034$), solute carrier family 13 member 4 (*Slc13a4*, $P = 0.000034$), aquaporin 1 (*Aqp1*, $P = 0.00047$), mitochondrially encoded ATP synthase membrane subunit 8 (*Mt-Atp8*, $P = 0.0015$), WD repeat domain 86 (*Wdr86*, $P = 0.0049$), protein phosphatase 1 regulatory subunit 17 (*Ppp1r17*, $P = 0.0023$), coagulation factor V (*F5*, $P = 0.029$), growth hormone (*Gh*, $P = 0.0028$), membrane frizzled-related protein (*Mfrp*, $P = 0.0085$), procollagen C-protease enhancer (*Pcolce*, $P = 0.012$), six transmembrane epithelial antigen of prostate 1 (*Steap1*, $P = 0.012$), cerebellin 3 precursor (*Cbln3*, $P = 0.022$), disabled-2 (*Dab2*, $P = 0.034$), FAT atypical cadherin 2 (*Fat2*, $P = 0.040$), angiotensin-converting enzyme (*Ace*, $P = 0.041$), retinol dehydrogenase 5 (*Radh5*, $P = 0.046$), dynein regulatory complex subunit 7 (*Drc7*, $P = 0.046$), and insulin-like growth factor-binding protein-2 (*Igfbp2*, $P = 0.046$).

We performed gene enrichment analysis on the differentially expressed genes in the F1 female hippocampus in Enrichr to identify any gene sets or gene ontology pathways that may be significantly overrepresented. After correcting for false discovery (FDR < 0.05), gene ontology analysis in Enrichr revealed 8 significantly enriched gene ontology pathways related to cellular components and biological processes (Fig. 5C). However, in the literature, many of these genes, perhaps most notably *Prl* and *Igfbp2*, are found to be differentially regulated in the hippocampus of rodents exhibiting stress and anxiety-related behaviors[41–44].

In the F1 males, there were initially 29 differentially expressed genes (FDR < 0.05) found in the hippocampus of the F1 male offspring of SARS-CoV-2 infected sires but only two differentially expressed genes had a log$_2$fold-change > 1 (Fig. 5D). These significantly downregulated genes included vestigial-like family member 3 (*Vgll3*, $P = 0.004$) and dachshund family transcription factor 1 (*Dach1*, $P = 0.049$).

We also investigated hippocampal gene profiles of the RNA-microinjected mice to see if there were any similarities in gene expression between RNA-microinjected and F1 mice. Due to the clustering of samples according to sex in the multidimensional scaling plot (MDS) analysis (Supplementary Fig. 6B), we performed the differential expression analyses separately for the RNA-microinjected males and females. Contrastingly, there was only one differentially expressed gene detected in the hippocampus of SARS-microinjected male mice (hypoxia-inducible factor 3α, *Hif3a*, $P = 0.026$) (Fig. 5E) and no differentially expressed genes detected in the RNA-microinjected female cohort (Fig. 5F). This suggests that the role of sperm small RNAs in altering the adult hippocampal transcriptome is likely to be limited.

## Discussion

Our previous studies have shown that both paternal viral-like immune activation and *Toxoplasmosis gondii* infection can have harmful multigenerational consequences for brain function and behavior and can change the sperm small noncoding RNA content[21,30]. Since SARS-CoV-2, a single-stranded RNA coronavirus, has infected hundreds of millions of men globally in the COVID-19 pandemic and continues to reinfect, there is a strong impetus to investigate whether paternal pre-conceptual SARS-CoV-2 infection also affects the brain and behavior of future generations. Using a mouse model of SARS-CoV-2 which displays the typical hallmarks of a moderate-to-severe human SARS-CoV-2 infection[33], our study demonstrates that pre-conceptual SARS-CoV-2 infection in male mice impacts the development and behavior of their offspring. Specifically, we saw increases in anxiety-like behavior of the adult F1 offspring (both sexes) from infected sires, with a slightly different profile seen in the male F1 offspring. We also observed additional F1 offspring phenotypic changes such as altered bodyweight development and decreased bodyweight changes after fasting, suggesting that other aspects of offspring physiology may be affected by paternal SARS-CoV-2. Furthermore, there were differences in the hippocampus transcriptome of the F1 female offspring, as well as subtle hippocampal transcriptomic changes in the male offspring. Additionally, we observed significant changes in the litter sizes and early-life bodyweight of the F2 grand-offspring of infected males, suggesting

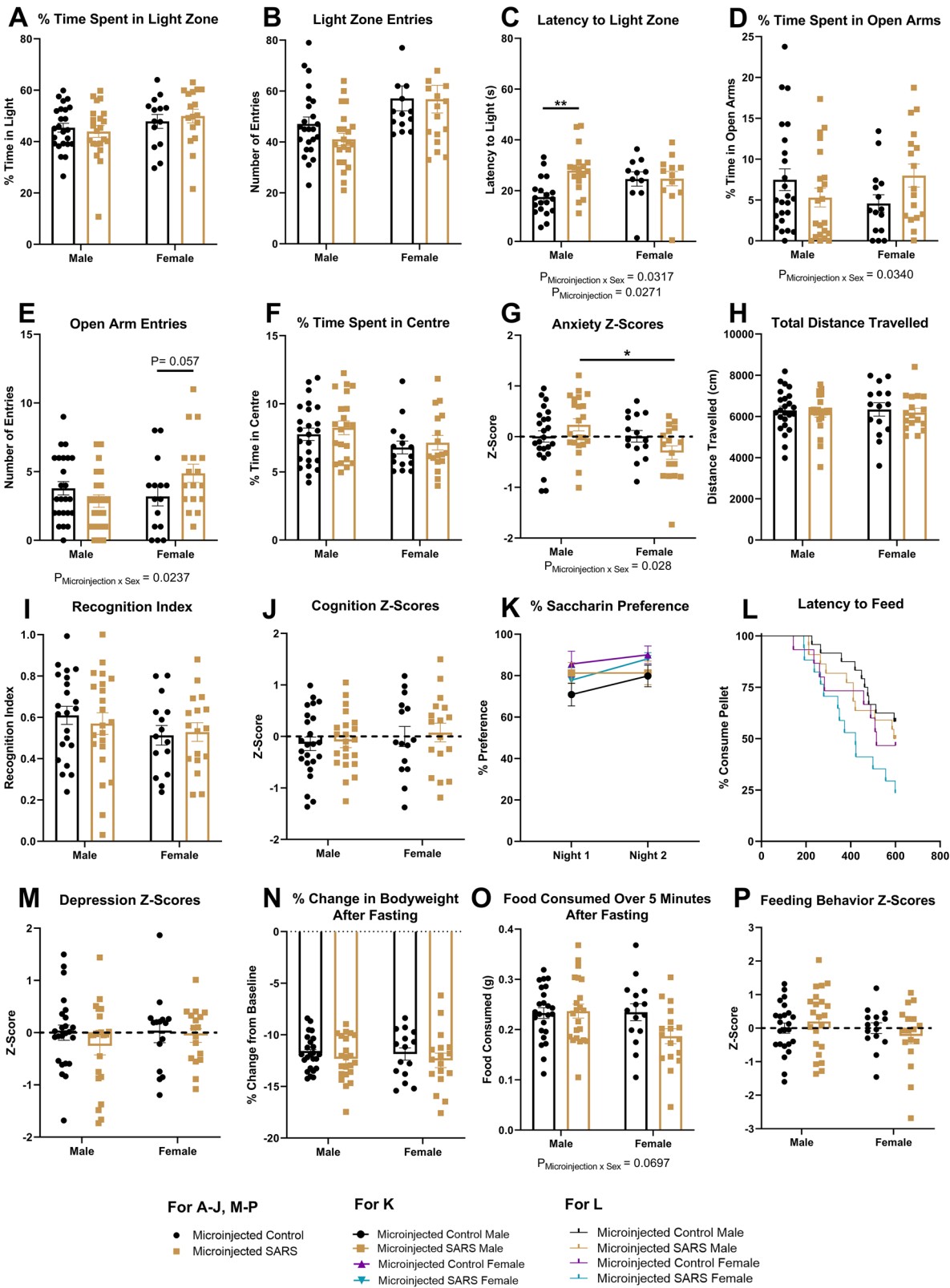

For A-J, M-P
- ● Microinjected Control
- ■ Microinjected SARS

For K
- ● Microinjected Control Male
- ■ Microinjected SARS Male
- ▲ Microinjected Control Female
- ▼ Microinjected SARS Female

For L
- ┼ Microinjected Control Male
- ┼ Microinjected SARS Male
- ┼ Microinjected Control Female
- ┼ Microinjected SARS Female

that paternal SARS-CoV-2 infection may have subtle developmental consequences beyond the next generation. Importantly, we identified changes in the sperm small noncoding RNA profiles of SARS-CoV-2 infected mice at the time of conception, showing that this molecular cargo of the sperm is altered by SARS-CoV-2. Through the microinjection of sperm RNA isolated from infected sires into naïve fertilized oocytes, we also have shown that these RNAs are functional in our paternal SARS-CoV-2 model. These findings add to the growing evidence that paternal environmental insults can reprogram offspring psychiatric phenotypes[16,45] and may thus provide insight into the developmental origins of psychiatric illnesses.

In both the light-dark box and open-field test, we saw that the paternal SARS-CoV-2 F1 male and female offspring mice spent significantly less time in the anxiogenic light zone and centre zone

**Fig. 4 | Microinjection of sperm RNAs from SARS-CoV-2 infected sires into fertilized oocytes significantly changes anxiety-like behavior in the resultant offspring.** Microinjected-SARS mice show no changes in (**A**) % time spent in the light zone of the light-dark box ($n = 24$ MCON M, $n = 14$ MCON F, $n = 22$ MSARS M, $n = 17$ MSARS F), or (**B**) number of entries into the light zone ($n = 24$ MCON M, $n = 14$ MCON F, $n = 22$ MSARS M, $n = 17$ MSARS F). There is a significant increase in (**C**) the latency to enter the light zone in the male microinjected-SARS mice only ($n = 19$ MCON M, $n = 11$ MCON F, $n = 18$ MSARS M, $n = 12$ MSARS (**F**)($P = 0.001$, general linear model with Bonferroni correction), and significant microinjection x sex interactions in the (**D**) % time spent in the open arms ($n = 24$ MCON M, $n = 15$ MCON F, $n = 22$ MSARS M, $n = 17$ MSARS F) and (**E**) number of entries into the open arms ($n = 24$ MCON M, $n = 15$ MCON F, $n = 22$ MSARS M, $n = 17$ MSARS (**F**)) of the elevated-plus maze. No differences are seen in the (**F**) % time spent in the centre of the open-field ($n = 24$ MCON M, $n = 15$ MCON F, $n = 21$ MSARS M, $n = 17$ MSARS F). There is a significant microinjection x sex interaction for (**G**) overall anxiety behavioral z-scores ($n = 24$ MCON M, $n = 15$ MCON F, $n = 22$ MSARS M, $n = 17$ MSARS F) ($P = 0.0132$, general linear model with Bonferroni correction). No differences seen in **H**) total distance traveled in the open-field ($n = 24$ MCON M, $n = 15$ MCON F, $n = 21$ MSARS M, $n = 17$ MSARS F), **I** the recognition index in trial 2 of the novel object recognition test ($n = 22$ MCON M, $n = 15$ MCON F, $n = 22$ MSARS M, $n = 17$ MSARS F), (**J**) overall cognition behavioral z-scores ($n = 24$ MCON M, n = 15 MCON F, $n = 22$ MSARS M, $n = 17$ MSARS F), **K** % preference for saccharin in the saccharin-preference test ($n = 24$ MCON M, $n = 15$ MCON F, $n = 22$ MSARS M, $n = 17$ MSARS F), (**L**) latency to feed in the novelty-suppressed feeding test ($n = 24$ MCON M, $n = 15$ MCON F, $n = 22$ MSARS M, $n = 17$ MSARS F), (**M**) overall depression behavioral z-scores ($n = 24$ MCON M, $n = 15$ MCON F, $n = 22$ MSARS M, $n = 17$ MSARS F), and (**N**) % bodyweight changed after a 24-h fasting period ($n = 24$ MCON M, $n = 15$ MCON F, $n = 22$ MSARS M, $n = 17$ MSARS F). There was a trend towards a significant micro-injection x sex interaction for **O**) food consumed in a 5-min period after a 24-h fasting period ($n = 24$ MCON M, $n = 15$ MCON F, $n = 22$ MSARS M, $n = 17$ MSARS F), but there were no differences in (**P**) overall feeding behavioral z-scores ($n = 24$ MCON M, $n = 15$ MCON F, $n = 22$ MSARS M, $n = 17$ MSARS F). Data presented as mean ± SEM. General linear models were used with *post-hoc* analyses where appropriate (Bonferroni-Holm corrected) except for (**L**). Cox regression with proportional hazards was used to analyse (**L**). Each n number refers to the number of individual animals per group. $P_{Microinjection}$ = main effect of microinjection treatment. $P_{Microinjection \times Sex}$ = interaction effect of microinjection treatment by sex. *$P < 0.05$, **$P < 0.01$.

respectively. These results indicate that SARS-CoV-2 infection in sires can increase overall anxiety-like behaviors in their offspring. Additionally, only the male offspring of infected sires took significantly longer to enter the light zone in the original F1 cohort, suggesting that these male offspring may initially have heightened perceived risk aversion (an anxiety-like behavior) in this test. Such sexual dimorphism is often observed in paternal non-Mendelian inheritance studies, albeit the underlying mechanisms are not yet clear[46]. Interestingly, this anxiety phenotype in the offspring is not congruent with the increased depression-like phenotype we observed previously in the offspring of the paternal viral-like immune challenge (Poly I:C; a viral mimic but not an infectious agent) model[30]. This suggests that although Poly I:C (via intraperitoneal injection) may mimic the acute immune-activation phase of a viral infection, it is likely that SARS-CoV-2 infection (initiated in the respiratory system and leading to symptoms modeling COVID-19) is affecting the next generation in a distinct manner. Indeed, unlike the acute but fairly mild sickness responses observed in the previously published Poly I:C model[30], our P21 SARS-CoV-2 model is characterized by more severe bodyweight loss over a longer period of time (7 days), cytokine storm and extensive lung inflammation[33], which may be leading to differential intergenerational effects in each model. On that note, a limitation in our model and other paternal infection models is that the bodyweight loss observed in our infected fathers may be contributing to the intergenerational changes seen, since paternal metabolic state on its own can have intergenerational health consequences[7,47,48]. However, since robust inflammation in mice is often associated with metabolic changes[49], this limitation cannot easily be avoided.

Alongside the increased anxiety observed in the F1 offspring, we identified 19 significantly downregulated genes and one significantly upregulated gene in the hippocampus of the adult female offspring of SARS-CoV-2 infected sires. While the gene ontology and KEGG pathway analyses did not provide much insight into the mechanistic relationships between these genes, many of the downregulated genes detected here have been implicated in other rodent models of stress and affective disorders[41,42,44]. For example, Stankiewicz and colleagues (2015) showed that *Aqp1*, *Prl*, and *Col8a2* were all downregulated in the hippocampus after acute social stress in mice[42]. Additionally, *Prl*, *F5*, *Otx2*, and *Aqp1* were also found to have reduced expression in the hippocampus of rodents displaying a heightened acute stress response compared to low-stress responders[41]. Furthermore, exposing mice to the stress of a fear-conditioning assay leads to the suppression of hippocampal genes including *Aqp1*, *Prl*, *Col8a1*, *Otx2*[50]. Taken together, these studies indicate that the downregulated genes in the hippocampus we have observed here have known links to anxiety and stress-related phenotypes. In particular, the prolactin system overall has been directly associated with anxiety in rodents, since inhibition of *Prl* in the brain increases anxiety-like behavior in the elevated-plus maze[51,52]. However, since very few differences were seen in the male offspring, who show a robust anxiety-like behavioral response alongside the females, it is highly likely that molecular and cellular changes in other brain regions are associated with the anxiety-like behaviors present in both the F1 male and female mice.

In our study, very few changes were observed in the F2 generation, suggesting that the transgenerational behavioral consequences of SARS-CoV-2 may be limited. Although changes in the F2 early-life bodyweight and litter sizes were evident, these did not translate into any overt and noteworthy phenotypic differences observed in adulthood. It is important to note the constraints of implementing behavioral neuroscience methods in a PC3 animal facility, since every behavioral assessment (and all mouse handling) must be performed in a size-restricted class II biosafety cabinet. Therefore, tests such as the Morris water maze, Y-maze, elevated plus maze, fear conditioning and other operant conditioning tests could not be adapted to fit inside the cabinet. It is therefore possible that we were unable to detect overt phenotypic changes in F2 (and possibly F1) due to these limitations. Nonetheless, it is rarely the case that the phenotypic outcomes of the F2 generation resemble that of the F1 generation in paternal epigenetic inheritance studies[30,45]. This is likely due to the different environmental (viral) and whole-body experiences faced by SARS-CoV-2 infected sires and F1 male breeders. The sires (F0) have experienced an infection whereas the F1 male breeders were never directly exposed to SARS-CoV-2 and associated symptoms modeling COVID-19.

Small noncoding RNAs, such as miRNAs, have been suggested to play a role in embryonic development[36], and there is growing evidence that the sperm delivers small noncoding RNAs that may have the ability to modulate offspring development[14,17,53]. Other studies have identified the epididymis as a region where the small noncoding RNA content of the maturing sperm can be modified, thus revealing a mechanistic pathway by which the host environment can alter the sperm RNA payload[36,39,54,55]. We observed that 2 small noncoding RNAs and 4 piRNA clusters were significantly altered in the sperm of SARS-CoV-2 infected sires, including miR-3471 and pro-TGG-3-1. Notably, none of these differentially expressed miRNAs overlap with the changes observed in the paternal Poly I:C and the paternal *T. Gondii* infection models, which reinforces the notion that each infection and immune activation model has a unique set of underlying mechanisms[21,30]. At this stage, the biological factors contributing to these differences between the sperm small noncoding RNA profiles of each infection and immune activation model are not well understood and require further

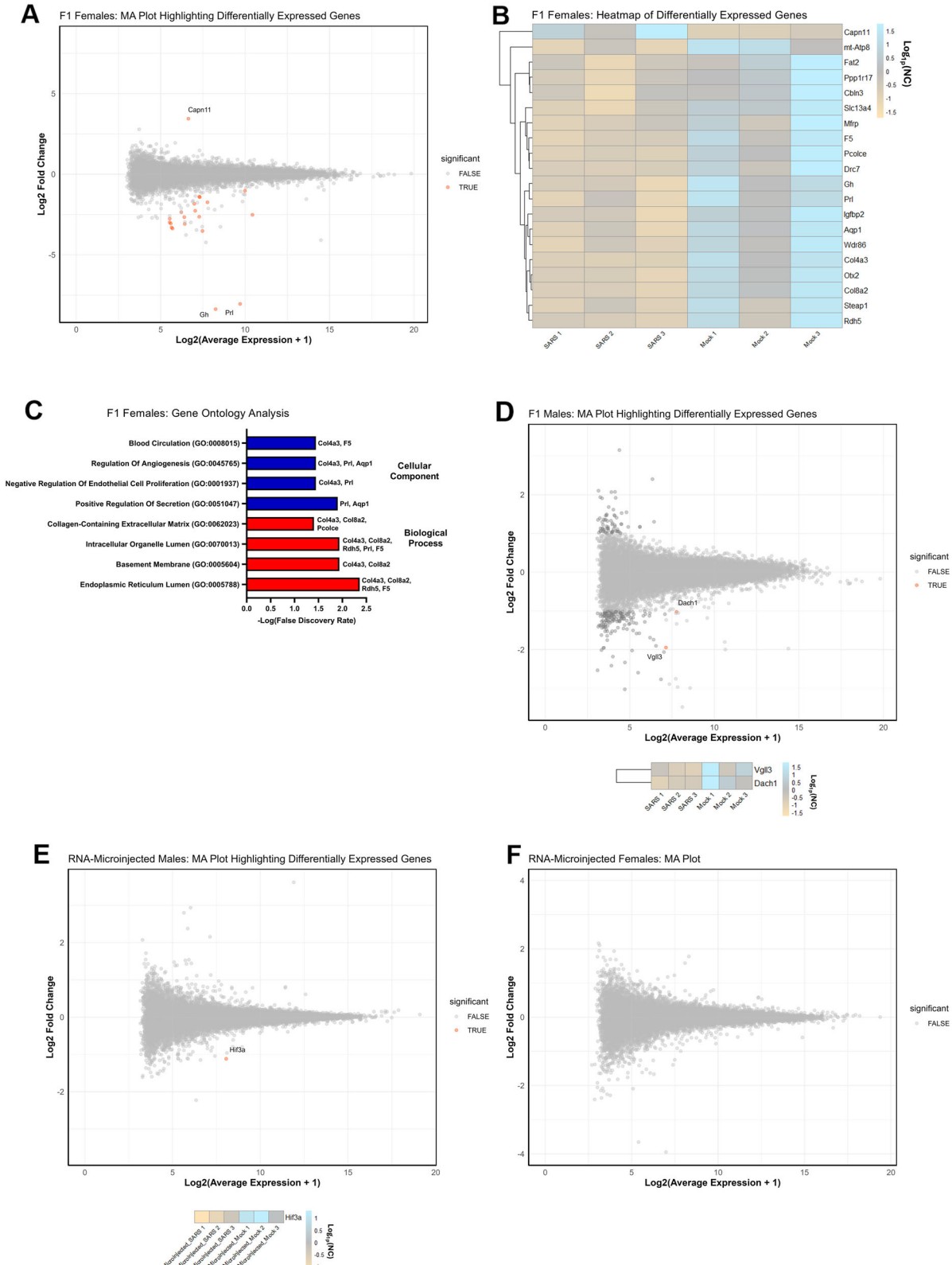

investigation. Whilst the timing between the paternal infection (or immune activation) event and conception is comparable between each of these paternal models (approximately 1 month), the severity of infection, the host immune profile, and the type of pathogen (e.g., virus or parasite) may all be contributing to the distinct sperm small noncoding RNA profiles observed in each paternal immune activation (PIA) model[21,29,30].

Importantly, our sperm RNA-microinjection into fertilized oocytes experiment reveals that the differentially expressed sperm RNAs from SARS-CoV-2 infected male mice may be functional in our paternal SARS-CoV-2 paradigm. This is seen particularly in the microinjected-SARS male mice that display increased anxiety-like behavior by taking longer to enter the anxiogenic light zone of the light-dark box. There is also evidence that anxiety-like behavior in the

**Fig. 5 | Paternal SARS-CoV-2 significantly alters the transcriptome of the hippocampus in F1 females and males. A** F1 female MA (Bland-Altman) plot of the $Log_2$(average expression + 1) for each gene vs. $Log_2$ fold-change relative to control values for each gene (red dots indicate genes which are FDR < 0.05), (**B**) Heatmap displaying the relative expression of the differentially expressed genes (FDR < 0.05) across all F1 female samples ($n = 3$ per paternal treatment group), (**C**) F1 female gene ontology pathways (y-axis) that were identified as significantly affected in the gene enrichment analysis of differentially expressed genes (gene symbols at the end of the bars indicating which gene/s are annotated to each term), (**D**) F1 male MA (Bland-Altman) plot of the $Log_2$(average expression + 1) for each gene vs. $Log_2$ fold-change relative to control values for each gene with a heatmap of differentially expressed genes below, (**E**) RNA-microinjected male MA (Bland-Altman) plot of the $Log_2$(average expression + 1) for each gene vs. $Log_2$ fold-change relative to control values for each gene with a heatmap of differentially expressed genes below, (**F**) RNA-microinjected female MA (Bland-Altman) plot of the $Log_2$(average expression + 1) for each gene vs. $Log_2$ fold-change relative to control values for each gene.

elevated-plus maze is globally changed in a sex-dependent manner by the microinjection of SARS-CoV-2 sperm RNAs; however, the *post-hoc* analysis does not clearly show that microinjected-SARS males (or females) are more anxious than their control-injected counterparts in this test. Not all F1 anxiety phenotypic changes are recapitulated by microinjection of sperm RNAs, especially in the light-dark box and open-field, which highlights the possibility that other biological mechanisms may be contributing to these behavioral changes. Nonetheless, our new data corroborates and significantly extends on the current literature showing that sperm-derived RNAs potentially play a role in the development of offspring phenotypes observed in different paternal intergenerational inheritance paradigms[14,15,18,21].

Epidemiological evidence from birth registries shows that the previous 1918 pandemic caused by Spanish Influenza led to multigenerational impacts on educational attainment, disability, and life expectancy[56]. However, since we are able to minimize genetic and environmental confounding factors in our preclinical SARS-CoV-2 study by using congenic male mice that do not participate in parenting, we demonstrate that SARS-CoV-2 infection in sires can increase anxiety levels in their offspring via changes to the molecular cargo of the sperm. Importantly, we have shown that SARS-CoV-2 related sperm RNA changes are at least in part contributing to the anxiety-related behaviors we observed in our natural F1 offspring. Future studies will include investigation of how sperm RNAs delivered to the oocyte modulate development and brain function in offspring.

Using well-designed cohort studies, it will be important to determine whether similar mental health outcomes can occur in the human offspring of sires exposed to a SARS-CoV-2 infection. However, unlike this preclinical study, undertaking these cohort studies with humans could potentially take decades. It will also be crucial to determine how the severity and timing of SARS-CoV-2 infection can influence the intergenerational outcomes. If undesirable intergenerational consequences can be avoided, provided a man conceives after a longer time period has elapsed from the time of infection, the best clinical approach may be to delay conception until the risk of any SARS-CoV-2 related intergenerational effects have dissipated. Another question raised from this study is whether prior exposure to a SARS-CoV-2 vaccine before infection or administration of antivirals early during acute infection can avoid or reduce the impact of SARS-CoV-2 infection on the sperm RNA payload. Investigating these questions further will be vital for mitigating the potentially serious public health implications arising from our study.

Overall, this study demonstrates that paternal exposure to SARS-CoV-2 infection leads to intergenerational changes in anxiety, hippocampal gene expression, and bodyweight development. We also show that SARS-CoV-2 infection can alter the sperm small noncoding RNA content at the time of conception and that these differentially expressed sperm-derived RNAs may partially be involved in the anxiety-related traits we have observed in the offspring. Furthermore, SARS-CoV-2 infection can lead to subtle changes in the early-life bodyweight of the grand-offspring, highlighting that SARS-CoV-2 may have subtle transgenerational consequences for health and development. Our study also suggests that paternal pre-conceptual viral infection may be an important determinant of mental health in the next generation. The present study extends on previous reports showing that paternal infection and inflammation can reprogram

offspring phenotypes. Given the large-scale global impact of the COVID-19 pandemic, it will be crucial to follow up these findings urgently with studies to determine whether these intergenerational impacts extend to humans. With this knowledge, we may be able to prevent or at least mitigate a potential surge of health problems, such as anxiety disorders, manifesting on a global scale in the next generation.

## Methods

### Intergenerational (F1) and transgenerational (F2) natural mating studies

**Subject details.** All mouse strains and experiments were reviewed and approved by the WEHI Animal Ethics Committee (AEC; approved application: 2020.016). Experiments were performed in accordance with the research guidelines and regulations of the National Health and Medical Research Council (NHMRC). Male and female C57BL/6 J mice were bred and maintained at the Walter and Eliza Hall Institute of Medical Research (WEHI). All procedures involving animals infected with SARS-CoV-2 or their progeny were conducted in the Physical Containment Level 3 (PC3) facility at WEHI (Cert-3621). Mice were transferred to the PC3 animal facility for all SARS-CoV-2 infection experiments or for breeding at least 1 week before the start of experiments to acclimatize to the room. Male and female wildtype mice were 8 weeks old at the onset of experiments. Mice were group housed (each sex and treatment group housed separately) with 4-6 mice per cage in individually ventilated microisolator cages (IVC) under level 3 biological containment conditions with a 12-h light/dark cycle (light on at: 07:00). Mice were provided with WEHI mouse breeder cubes (Ridley Agri Products) and sterile acidified water *ad libitum*.

**SARS-CoV-2 strain and murine infection.** The P21 SARS-CoV-2 mouse-adapted virus used in this study has been previously described[33]. For paternal infection, 8-week-old male C57BL/6J mice were anesthetized with methoxyflurane and received intranasal instillation of SARS-CoV-2 P21 isolate ($10^4$ TCID50 in 30 μl) or PBS (mock infected control). After infection, mice were physically checked and weighed daily for a period of 10 days, after which time the mice were visually checked daily. In this model, the infected mice typically show peak viral loads of $10^9$ TCID50/lung at day 2 post-infection and clear the infection by day 7 to 10 post-infection[33]. Mice also typically lose approximately 9–15% bodyweight by day 3 post-infection, which is a strong correlate of the severity of infection[33]. All SARS-CoV-2 P21 infected male mice used in this study reached their peak weight loss at day 3 post-infection. In contrast, and as expected, mock-infected control mice did not lose a significant amount of weight following PBS treatment and anesthesia (Supplementary Fig. 1A). To ensure we were only mating mice with a moderate-to-severe infection in the SARS-CoV-2 treatment group, we mated those that lost between 9–15% of their bodyweight at day 3 post-infection.

**Mating.** Four weeks after infection, male mice (12 weeks of age) that were either infected with SARS-CoV-2 and lost at least 9% of their bodyweight at the peak of infection, or were mock infected (controls), were mated in trios with naïve female mice (8 weeks of age). After a 6-day mating period, the male mice were removed and the female mice

were separated and single-housed until they had littered. To generate grand-offspring (F2 mice), 10-week-old F1 male offspring from either SARS-CoV-2 infected or control sires were mated with naïve 8-week-old female C57BL/6J mice using the protocol described above. Male mice used for breeding were euthanized via $CO_2$ asphyxiation within three days of mating.

In a separate cohort, blood was collected from both male and female mice after the mating period through cardiac puncture and stored in tubes containing EDTA. Following centrifugation at 1100 $g$ for 15 min, plasma was collected and stored at −80 °C until an ACROBiosystems SARS-CoV-2 spike protein serological IgG ELISA (cat no. RP-13) was performed according to manufacturer's instructions. This confirmed that there were no IgG antibodies against SARS-CoV-2 spike protein in any of the female breeders while there was robust expression of IgG antibodies in the male breeders previously infected with SARS-CoV-2 (Refer to Source Data). Fecal pellets were also collected from the female breeders after the mating period (post-mortem tissue dissection from the rectal area) to investigate any potential gut microbiota changes. Fecal samples were then stored at −80 °C until DNA extraction for 16S sequencing and analysis. The left testis was dissected from the male mice in this cohort and the tunica albuginea (capsule) was pierced 20 times with a fine needle before immersing the testis in 10% neutral buffered formalin for 72 h, followed by immersion in 70% ethanol for 24 h.

**Testes embedding, sectioning and staining.** After alcohol processing, each testis was cut in half transversally with a razor and then embedded in paraffin. 5μm sections were cut from the middle portion of each testis. Sections were dewaxed in xylene (3 × 5 min) then rehydrated in ethanol (3 × 2 min in 100% ethanol followed by 5 min in 70% ethanol) and then washed with distilled water before proceeding with staining.

Sections were stained with periodic acid Schiff (PAS) reagent and haematoxylin. Firstly, periodic acid was applied for 10 min followed by a 5-min wash with distilled water. Then, Schiff's reagent was applied for 15 min followed by a 5-min wash with distilled water. Meyer's haematoxylin was applied for 20 s followed by a 5 min-wash with distilled water. Finally, Scott's tap water was applied for 5 min followed by a 5 min wash with distilled water. Stained sections were then dehydrated with ethanol (5 min in 70% ethanol followed by 3 × 2 min in 100% ethanol) and then xylene (3 × 5 min).

**Testes histological examination.** After PAS and haematoxylin staining, sections were scanned at 20X magnification using an Olympus BX53 microscope, fitted with an Olympus DP80 digital camera. Seminiferous tubules were examined for presence of the various germ cell types. Seminiferous tubules in stage II and stage III of spermatogenesis were identified on three independent sections per animal. Spermatocytes, round spermatids, and elongated spermatids were identified and manually counted by a blinded, experienced researcher. These counts for each cell type were averaged across the three tubules per animal and subsequently analyzed.

**Fecal DNA extraction and quantification.** Fecal pellets from female breeders were mechanically and chemically lysed, as previously described[57], using the Qiagen QIAamp PowerFecal Pro DNA Kit (Cat. # 51804) for DNA extraction according to the manufacturer's instructions. Samples were then quantified for DNA concentration using the NanoDrop Ultra Spectrophotometer (ThermoFisher Scientific).

**16S Sequencing.** We performed full length 16S rRNA sequencing of the fecal DNA samples using PacBio HiFi Technology by the Australian Genomic Research Facility (AGRF, Melbourne, Australia). Sequencing data were quality filtered and denoised to generate high-resolution amplicon sequence variants (ASVs) using QIIME2[58] in conjunction with the DADA2 pipeline[59]. Taxonomic classification of ASVs was performed using two complementary approaches. First, a consensus-based alignment was performed with VSEARCH against the Genome Taxonomy Database (GTDB, release 207). Second, a naïve Bayesian classifier (DADA2) was applied using a tiered database approach: classification was attempted sequentially using the GTDB (r207), the SILVA rRNA database (v138), and finally the NCBI RefSeq 16S rRNA database supplemented with the Ribosomal Database Project (RDP). This tiered strategy improves classification accuracy, particularly for low-abundance ASVs. Alpha diversity was assessed using the Shannon diversity index, which quantifies both species richness (number of taxa present) and evenness (relative abundance distribution).

**F1/F2 experimental timelines.** In the F1 generation, there were 10 litters in the paternal SARS-CoV-2 group and 9 litters in the paternal control group. In the F2 generation, there were 7 litters for the grand-paternal SARS-CoV-2 group and 6 litters for the grand-paternal control group. Prior to weaning, the pups were weighed on post-natal day (PND) 8, PND 15, and PND 22. At weaning (3 weeks of age), the F1/F2 offspring were divided into cohorts and housed according to sex and treatment group (n = 4–6 per cage) with cage mates from other litters within the same treatment to mitigate litter effects. Handling of the F1/F2 offspring was kept to a minimum other than weekly cage changes and bodyweight measurements until 8 weeks of age. At 8 weeks of age, behavioral testing commenced in the behavioral F1/F2 cohorts. Separate cohorts that were not exposed to behavioral testing, including one cohort for breeding of F2 and one cohort for tissue collection (both F1 and F2), were left undisturbed until 10 weeks of age. The main behavioral battery was performed in a similar manner for the first cohorts in F1 and F2 (see Fig. 1A). Another independent F1 cohort (n = 4–10 per group) was used to repeat anxiety-like behavioral tests, including the light-dark box and open-field test, and data from the two F1 cohorts for the anxiety tests was combined into one analysis. To ensure reproducibility, we performed the main behavioral battery in a separate fully powered F1 cohort and have presented these results in Supplementary Fig. 3. Tests in each battery were performed in order of the least stressful to the most stressful for the mice, with at least one day rest in-between each test. All tests were performed during the light phase (7:00–19:00) within a class II biosafety cabinet to comply with PC3 facility procedures and regulatory requirements. For the light-dark box, open-field, novel object recognition test, social interaction test and novelty-suppressed feeding test, each mouse was video recorded using an overhead camera. Other than the social interaction test and novelty-suppressed feeding test, TopScan Lite (Cleversys Inc.) was used to analyse the videos. The class II biosafety cabinet was covered with a screen to control light levels and reduce visual distractions. Experimenters were blinded to treatment groups during testing and analysis phases.

**Light/dark box.** This test was used to assess anxiety-like behavior[30,60]. The apparatus consisted of a clear Perspex box (40 × 40 cm) and a black Perspex insert with an open archway at the bottom to allow free movement of the mice. With the insert, the box was equally divided into a dark hidden zone and a brightly lit light zone (750 lux). At the start of each trial, the mouse was individually placed in the dark zone. The mouse was allowed roam freely in the light-dark box for 10 min while being video recorded. The percentage duration spent in the light compartment, the latency to enter the light compartment, and the number of entries made into the light compartment were calculated and compared between treatment groups as indicators of anxiety-like behavior.

**Open-field test.** The open-field test was used to evaluate anxiety-like behavior and locomotion according to a previously published protocol[21]. Briefly, the mouse was initially placed in the perimeter zone

of an open arena (33 × 33 cm, 50 lux), which contained a defined centre zone (15 × 15 cm) and perimeter zone. The mouse was allowed to roam freely for 10 min while being video recorded. The total distance traveled, and the percentage time spent in the centre zone were calculated.

**Novel-object recognition test.** The novel-object recognition test was performed to assess short-term memory according to our previously published protocol[7]. Mice were habituated to the empty testing arena (33 × 33 × 27 cm plastic box) for 10 min at 24 h prior to trial 1. For objects, we used a 50 ml Falcon tube filled with woodchip bedding and a tower made of LEGO bricks of an equivalent height. During the first trial, the mice were reintroduced to the testing arena where two identical objects were located and allowed to freely explore the arena and objects for 10 min. A second trial was performed after an inter-trial interval of 1 h. For the second trial, mice were individually placed into the testing arena, which now contained a familiar object (from trial 1) and a novel object. The assignment of familiar and novel objects was randomized and balanced across treatment groups. Mice explored the arena and objects for 5 min in trial 2 while being video recorded. The recognition index from trial 2 was calculated as a fraction of time exploring the novel object over total time spent exploring both objects.

**Social interaction test.** We investigated social interaction abilities and interest according to a previously published protocol[61]. The mouse (test mouse) was placed in the arena (33 × 33 cm) and then a guest mouse was introduced. This guest mouse was age, sex and weight matched, had been group housed, but had never previously encountered the test mouse. The mice were allowed to freely explore the arena and each other for 10 min while being video recorded. Video recordings were later scored by a blinded experimenter using SocialScan (Cleversys Inc.). Total contact time with the guest mouse (within 2 cm of each other) was calculated for each test mouse.

**Novelty-suppressed feeding test.** To measure anxiety-related depression and hyponeophagia, we performed the novelty-suppressed feeding test according to a previously published protocol[62]. The mice were deprived of food for 24 h prior to testing. Bodyweight was recorded pre- and post-fast. The apparatus for testing was a large white plastic container (64.5 × 41.3 cm) lined with 2 cm of woodchip bedding material and a standard chow food pellet placed in the centre (750 lux) on a piece of filter paper. The mouse was initially placed on the side of the arena and allowed to roam freely. The latency to reach a stationary position while feeding on the chow pellet was recorded (maximum of 10 min). After this, the mouse was removed from the arena and their individual food intake over 5 min was measured to evaluate any potential differences in hunger drive between treatment groups that may confound this test.

**Sucrose preference test.** The sucrose preference test, which measures anhedonia, was adapted from a previously published protocol[63]. After a 24-h habituation period to the two-bottle set up (normal drinking water), the mice were habituated to the 1% sucrose solution (Sigma Aldrich) in their home cages for 24 h. On day 3, mice were exposed to one bottle containing 1% sucrose and the other containing normal drinking water for 24 h. On day 4 at 5 pm, drinking water and food were removed from the cages so that the mice were food and water deprived for 18 h. On day 5 (testing day), mice were individually housed for 8 h and given a choice between 1% sucrose and normal drinking water. Food intake was also measured over this period. The weight of the bottles before and after the 8-h testing period were measured and used for the calculation of sucrose preference. Sucrose preference was calculated as a proportion of sucrose consumption over the total fluid consumed.

**Behavioral Z-scoring.** To provide comprehensive and integrated behavioral analyses across the multiple tests performed, a normalization approach known as behavioral z-scoring was used according to methodology previously described[64,65]. Firstly, for each behavioral parameter, z-scores were calculated for each individual mouse according to the formula below. This formula measures how many standard deviations (σ) a data point (x) is above or below the mean (μ) of the control group (in this case, the paternal mock infected group or grand-paternal mock infected group).

$$Z = \frac{x - \mu}{\sigma}$$

The directionality of each Z-score was inverted in the cases where an increase in the Z-score for a particular parameter translated to a decrease in the behavioral trait it represents. Z-scores were then averaged within a behavioral measure and then across all behavioral measures within a test (e.g entries into light zone, latency to light zone and % time spent in light zone for the light/dark box). Z-scores were then averaged across related tests to give an integrated behavioral score for each behavioral domain. The main behavioral domains included "anxiety", "depression", "sociability" (F1 only), "cognition", and "feeding". The table below outlines which tests were included in each domain.

| Behavioral Z-Score Domain | Behavioral Tests |
| --- | --- |
| Anxiety | Light/dark box, Open Field, Elevated Plus Maze (Microinjected cohort only) |
| Depression | Sucrose Preference Test, Novelty-Suppressed Feeding Test |
| Cognition | Novel Object Recognition Test, Y-maze (Microinjected cohort only) |
| Sociability | Social Interaction Test (F1 only) |
| Feeding Behavior | Food Intake (Sucrose Preference Test), Food Intake (Novelty-Suppressed Feeding Test), and % Change in Bodyweight After Fasting (Novelty-Suppressed Feeding Test) |

**Immune challenge.** The first F1/F2 cohort were given a viral-like immune challenge in the form of a Poly I:C injection to assess their pro-inflammatory immune response. Poly I:C is a toll-like 3 receptor agonist that triggers the release of systemic pro-inflammatory cytokines such as IL-6, IL-1β, and TNF-α[66]. Poly I:C Potassium Salt (P9582) was obtained from Sigma Aldrich (lot number 12181209). The solution was prepared freshly on the day of injection by dissolving the Poly I:C in RNase-free water at room temperature to a concentration of 1 mg/ml. Mice were intraperitoneally injected with 12 mg/kg bodyweight Poly I:C or a 0.9% saline solution injection (control). At 2-h post-injection, the mice were euthanized via $CO_2$ asphyxiation and blood was collected through cardiac puncture and stored in tubes containing EDTA. Following centrifugation at 1100 $g$ for 15 min, plasma was collected and stored at −80 °C until an IL-6 ELISA was performed. Plasma IL-6 was measured in response to Poly I:C, as this cytokine is known to be acutely upregulated by Poly I:C at the 2-h interval post-injection[3]. 1% Triton-X-100 solution was added to plasma samples prior to removal from the PC3 facility. IL-6 concentrations were measured using a validated uncoated Mouse IL-6 ELISA kit (Cat. #88-7064: Invitrogen) according to manufacturer's instructions.

## Sperm small noncoding RNA analysis
**Mature spermatozoa collection.** In an independent cohort of mice, 8-week-old adult male mice were infected with SARS-CoV-2 or mock

infected with PBS (control). Four weeks after infection, which corresponds to the time of mating in the previous cohort, these male mice were culled to collect mature spermatozoa from the epididymides. To isolate mature spermatozoa, the caudae epididymides from each mouse were incised and placed in Eppendorf tubes containing Modified Tyrode's Medium 6 (MT6) (solution contains: 124 mM NaCl, 2.68 mM KCl, 17.14 mM CaCl$_2$, 3.2 mM NaH$_2$PO$_4$.H$_2$O, 0.49 mM MgCl$_2$.6H$_2$O, 25 mM NaHCO$_3$, 5.6 mM D-glucose, 4 mg/ml BSA, 28.2 mM Phenol Red) and incubated at 37 °C for 30 min. Following this 'swim-out' period, epididymides were then removed and after centrifugation at 400 $g$ for 10 min, the supernatant was removed. Sperm pellets were stored at −80 °C until the small RNA extraction. The testes (left and right together) were also weighed from this cohort.

**Sperm RNA extraction and small noncoding RNA Illumina sequencing.** Pooling of the sperm from three mice per biological replicate ($n$ = 4–5 biological replicates per treatment group) was conducted to increase RNA yield. 15 mice in the control group and 12 mice in the SARS-CoV-2 infected group were represented across all biological replicates. These mice were sampled equally from three independent cohorts. Total RNA containing small noncoding RNA were purified from samples using the QIAGEN miRNeasy Mini Kit (Cat. #217004) according to manufacturer's instructions. DNase treatment was also performed using the Invitrogen DNA-freeTM Kit (Cat. #AM1906) according to manufacturer's instructions. Quantification of total RNA in each sample was performed using the Qubit 4 Fluorometer and the Qubit RNA BR Assay Kit (Cat. #Q10210). Quality control of each RNA sample was conducted using the Agilent 4200 TapeStation System and the RNA ScreenTape, RNA ScreenTape Ladder, and RNA ScreenTape Buffer (Cat. #5067–5576 to #5067–5578). NEXTflex Small RNA v4 library preparation with bead size selection was performed followed by Illumina Next Generation sequencing using the Illumina NovaSeq 6000 SP workflow at AGRF. 300 ng of RNA per sample was used for library preparation. All samples were run on a single flow cell lane using 100-bp single-end reads. An average of 42,569,174 raw reads per biological replicate were obtained. After the deletion of the 3' adapter sequences and random bases, an average of 34,066,274 trimmed reads per biological replicate were obtained. One replicate from the SARS-CoV-2 infected group was removed due to poor quality RNA as determined by FastQC[67].

**Sperm miRNA sequencing analysis.** Quality control of the reads was performed using FastQC[67]. Cutadapt was used to trim sperm RNA reads against known NEXTflex Small RNA Sequencing Kit v4 3'adapters and random bases[68]. The Rsubread package (v 2.12.3) was used to generate an index consisting of mm10 miRNA sequences (miRBase Fasta format, URL: https://mirbase.org/ftp.shtml). Mapped reads were size restricted to include only those that were 19–26 nt in length (corresponds to miRNAs). Rsubread's featureCounts function was used to summarize all the successfully mapped reads into counts[69]. Annotation files from miRbase v 22.1 (miRNAs) were used to annotate the miRNAs.

Lowly expressed miRNAs were filtered out by excluding those with less than a count of 1 in 2 samples. The DESeq2 package was used to perform statistical analysis to identify significantly differentially expressed miRNAs[70]. The false discovery rate was controlled below 5% using the Benjamini and Hochberg method. A cut-off of Log$_2$ fold expression change of 1 and above (or −1 and below) relative to the control group was included in the analyses.

Differentially expressed miRNAs were applied to TargetScan[71] to look at predicted downstream gene targets. Predicted gene binding analysis was restricted to those that were above 80 predicted score and expressed in the early stages of embryogenesis. Functional annotation of these gene targets was conducted using Enrichr[40] to look at Kyoto Encyclopedia of Genes and Genomes (KEGG) pathways[72] and

gene ontology pathways that were significantly regulated by these genes (FDR < 0.05). The MA plots were generated using the R package ggplot (v 3.4.4) and the heatmaps were produced using the pheatmap package (v 1.0.12).

**Sperm piRNA cluster prediction and differential expression analysis.** Raw sequencing reads were subjected to trimming and quality control steps as performed above in miRNA analysis to ensure accurate downstream analysis. Reads were then collapsed into unique sequences to eliminate redundancies and facilitate more efficient mapping, utilizing the previously published collapse scripts in a Linux environment[73]. For the mapping of processed reads to the mouse reference genome (GRCm38, GENCODE released (https://www.gencodegenes.org/mouse/release_M25.html), we employed the Bowtie alignment tool[74], which is optimized for aligning short sequences to large genomes. The SAM files generated by Bowtie were subsequently used as input for the piRNA cluster prediction tool, proTRAC 2.4.3[37]. proTRAC was configured to perform a sliding window analysis with a window size of 5000 bp and an increment of 1000 bp. The tool applied statistical normalization based on the number of genomic hits and the total number of sequence reads to ensure that piRNA hit densities were accurately represented. Additionally, the tool set thresholds for piRNA-specific features including the typical piRNA length of 16–33 nt. Cluster prediction was validated by assessing the significance of hit densities within these genomic regions, with a significance threshold set at p ≤ 0.01. Post-prediction, the identified piRNA clusters were merged across all samples and then used to intersect gene annotation file (gencode.vM25.annotation, GENCODE released (https://www.gencodegenes.org/mouse/release_M25.html) for further functional study. Each sample (SAM) was then converted to BAM format and counted against the merged piRNA clusters to generate a count matrix using Bedtools[75]. This count matrix was subsequently used to prepare for differential expression analysis.

For differential expression analysis, read counts corresponding to each predicted piRNA cluster were generated. The DESeq2 package was used to perform statistical analysis to identify significantly differentially expressed piRNA clusters between experimental groups[70]. Low count reads were removed with the threshold of 50 hits in at least 4 samples. A cut-off of Log$_2$ fold expression change of 1 and above (or −1 and below) relative to the control group was included in the analyses.

**tsRNA identification, quantification, and differential expression analysis.** The systematic identification and characterization of tRNA-derived small RNAs (tsRNAs) were conducted using the TDRmapper[76]. The analysis commenced with quality control of raw small RNA sequencing data with FastQC to ensure the integrity of subsequential analytical processes[67].

The processed reads were then input to tDRmapper together with mouse reference genome (mm10 FASTA provided by tDRMapper, (https://github.com/sararselitsky/tDRmapper) for read filtration, alignment, annotation, and quantification[76]. Specifically, tDRmapper applied a comprehensive mapping strategy that discriminates between tRNA fragments and other RNA biotypes by considering exact matches, mismatches, and deletions to identify and annotate tRNA-derived fragments[76]. Following the quantification process, the output files from each sample were merged and used to generate a count matrix. Differential expression analysis was then conducted to explore variations in tsRNA profiles across SARS-CoV-2 and control groups. This analysis was performed using the DESeq2 package, which enabled the statistical assessment of expression changes of the identified tsRNAs between SARS-CoV-2 and mock infected (control) mice[70]. A cut-off of Log$_2$ fold expression change of 1 and above (or −1 and below) relative to the control group was included in the analyses.

**Predicted gene targets and gene ontology analyses.** The differentially expressed miRNA was applied to the miRDB database (https://mirdb.org/) to look at predicted downstream gene targets. Only genes expressed in early embryonic development with a target score of 90 and above were included in further functional annotation. The differentially expressed tsRNA was applied to TargetScanMouse Custom Release 5.2, which allowed us to enter custom small RNA sequences two to eight nucleotides in length (https://www.targetscan.org/mmu_50/seedmatch.html). Functional annotation of these gene targets was conducted using Enrichr[40] to look at gene ontology pathways and Kyoto Encyclopedia of Genes and Genomes (KEGG) pathways[72] that were significantly regulated by these genes (FDR < 0.05). The figures were generated by the R package ggplot (v 3.4.4) and the pheatmap package (v 1.0.12).

### Sperm RNA microinjection study

**Subject details.** All experiments were approved by the Florey Institute of Neuroscience and Mental Health Animal Ethics Committee (ethics approval number: 2024-084 FINMH) and were performed following the research guidelines and regulations of the National Health and Medical Research Council (NHMRC). C57BL/6J mice and fertilized oocytes from the WEHI breeding facility (Kew, Victoria, Australia) were used for the behavioral studies and microinjections (WEHI ethics approval number: 2023.003). Mice were weaned at 3 weeks of age (PND 21) and weighed weekly from 4 weeks until 12 weeks of age. After weaning, mice were transferred from WEHI to the Florey Institute of Neuroscience and Mental Health. Upon arrival from WEHI at 4 weeks of age, mice were group housed (each sex and treatment group housed separately) with 3–5 mice per open-top cage. All mice received standard chow and water *ad libitum* unless otherwise stated. The holding room was temperature controlled (22° Celsius, 45% humidity) with a 12:12 h light/dark cycle (lights on at 06:30). Mice were euthanized at 12 weeks of age via cervical dislocation.

**Small noncoding RNA purification and quality control for microinjections.** The total RNA samples extracted from mature spermatozoa of SARS-CoV-2 infected male mice (n = 4 samples with sperm RNA from 12 mice) and control male mice (n = 5 samples with sperm RNA from 15 mice) used for RNA sequencing in the sperm small noncoding RNA analysis were pooled into one sample per treatment group. Prior to pooling, quality control for each RNA sample was performed using the Agilent 4200 TapeStation System and the RNA ScreenTape, RNA ScreenTape Ladder, and RNA ScreenTape Buffer (Cat. #5067–5576 to #5067–5578). After pooling, the small RNA fraction in each pooled sample was enriched using the QIAGEN RNeasy Min Elute Cleanup Kit (Cat. #74204) as previously conducted in Tyebji et al. 2020. A graph showing the size distribution of RNAs found in a pooled sperm RNA control sample after small RNA enrichment has been provided in Supplementary Fig. 5A. Quantification of the small RNA in each sample was performed using the Qubit 4 Fluorometer and the Qubit RNA BR (broad range) Assay Kit (Cat. #Q10210). Each small RNA enriched solution was diluted to a concentration of 1 ng/μl of RNA using a Tris-EDTA microinjection buffer (100 mM Tris, 0.1 mM EDTA, pH 7.4), which is in accordance with previously published sperm small noncoding RNA microinjection studies[18,21].

**Oocyte microinjections.** The microinjections of the small noncoding RNA enriched solutions into naïve fertilized oocytes were performed by trained laboratory technicians at the WEHI Kew Facility (Ethics approval 2023.003). Prior to mating, C57BL/6J female mice were super-ovulated with subcutaneous injections of 5 IU of PMSG (Prospec) on day 1 and then 5 IU of hCG (Chorulon) on day 3. These mice were then immediately mated with C57BL/6J male mice and after one day, the embryos were collected. The RNA solution was microinjected into the embryos until the distension of the male pronucleus was observed, which was approximately 1–2 picolitres. Overall, 147 embryos were microinjected with RNA from the SARS-CoV-2 infected mice (microinjected-SARS) and 151 embryos were microinjected with RNA from the mock infected mice (microinjected-control). Of these, 109 microinjected-SARS 2-cell embryos were transferred to recipient females and 124 microinjected-control 2-cell embryos were transferred to recipient females. Following this, 45 microinjected-SARS pups and 43-microinjected control pups were littered. Overall, 41 microinjected-control and 39 microinjected-SARS mice survived beyond weaning. After transfer to the Florey Institute, these offspring were left undisturbed other than having weekly bodyweight measurements taken until 8 weeks of age when behavioral assessment started.

**Behavioral assessments.** At 8 weeks of age, behavioral testing commenced on the RNA-microinjected offspring. Tests were performed in an order similar to the 'main battery' in Fig. 1A. For all tests, the mice were habituated to the room where the behavioral assessment was occurring for at least 1 h. Experimenters were blinded to treatment groups during testing and analysis phases. All tests were conducted in a similar manner to the natural F1/F2 cohorts with the addition of the elevated plus maze (anxiety test following the open-field test) and the Y-maze (cognitive test following the novel object recognition test). These tests could not be performed in the natural cohorts due to the size constraints of the workspace in the PC3 facility. The saccharin preference test (anhedonia test) was performed instead of the sucrose preference test in the RNA-microinjected offspring.

**Light/dark box.** This test was conducted to assess anxiety-like behavior[30,60]. This test was conducted using the Med Associates ENV-510/511 open-field arenas (Fairfax, VT, USA) with a dark Plexiglas insert covering half of the arena. The light zone was 750 lux. Mice were initially placed in the dark zone and were allowed to roam freely in the arena for 10 min. Mice that immediately entered the light zone (< 0.1 s) at commencement of the test were excluded from latency to enter light zone data as this represented a startle response rather than risk evaluation. The percentage duration spent in each compartment, the latency to light zone, and the number of light zone entries were measured.

**Large open-field test.** The large open-field test was conducted to assess anxiety and locomotion[30]. The circular open-field arena was 1 m in diameter with a brightly lit centre zone (1000 lux) and a perimeter zone. Each mouse was initially placed in the centre zone and was then allowed to roam undisturbed in the arena for 10 min. The total distance traveled, and the percentage time spent in the centre zone were calculated.

**Additional tests: Elevated-plus maze.** The elevated-plus maze was performed to measure anxiety-like behavior according to our previously published protocol[30]. The apparatus formed a plus-shaped maze consisting of two open arms (5 cm × 30 cm) and two closed arms (5 cm × 30 cm) with a centre zone in the middle (5 cm × 5 cm). Each mouse was initially placed in the centre zone and allowed to roam freely for 5 min. The percentage time spent in the open arms and number of open arm entries were measured.

**Novel object recognition test.** The novel object recognition test was used to assess short-term memory and was performed with the same protocol as stated in the natural F1/F2 studies.

**Additional tests: Y-maze.** The Y-maze was conducted to assess spatial short-term memory[30]. The apparatus (San Diego Instruments, CA, USA) consisted of three arms (each 30 cm in length) at 120 degrees to each other which formed the shape of a Y. There were also visual symbols attached to the end of each arm to assist with spatial

orientation. Each mouse underwent two trials in the test with an inter-trial interval of 1 h. For the first trial, each mouse was initially placed in the 'home' arm and allowed to roam freely for 10 min in this arm and one other arm of the maze (familiar arm). The remaining arm (novel arm) was blocked off by a sliding door for the first trial. During the second trial, the mouse was placed back into the home arm and was allowed to roam freely in all three arms of the Y-maze for 5 min. Novel arm preference was calculated as a proportion of the time exploring the novel arm (s) over an average of the time spent exploring both the home arm and the familiar arm (s) for the last 4 min in trial 2.

**Novelty-suppressed feeding test.** The novelty-suppressed feeding test was performed with an almost identical protocol as in the natural F1/F2 studies. The main difference was that the test was conducted in a large square arena (60 cm × 60 cm × 60 cm) lined with 2 cm of wood-chip bedding material and a standard chow pellet placed in the centre on a piece of filter paper.

**Additional tests: Saccharin preference test.** The saccharin preference test was used to measure hedonic behavior and any deficits associated with hedonic behavior (anhedonia) commonly seen in depression according to our previously published protocol[30]. Firstly, mice were single housed and exposed to the two-bottle set-up (each containing water) for 24 h. On the second and third day (testing period), the mice were exposed to one bottle containing saccharin sweetened water (0.1%, Sigma Aldrich) and another bottle containing water. The arrangement of the bottles was randomized and switched between day 2 and day 3 of the test. Saccharin and water consumption were measured each morning. Saccharin preference was calculated as a percentage of saccharin consumption over total fluid consumption.

**RNA sequencing of the F1 and RNA-microinjected hippocampus**
**Hippocampus dissection, total RNA extraction and Illumina mRNA sequencing.** Both the left and right whole hippocampus were dissected from the F1 and RNA-microinjected cohorts used for tissue collection at 10–12 weeks of age, snap frozen, and stored at −80 °C until the RNA extraction was performed.

Total RNA was purified from the hippocampus samples using the QIAGEN miRNeasy Mini Kit (Cat. #217004) according to manufacturer's instructions. DNase treatment was also conducted using the Invitrogen DNA-$free^{TM}$ Kit (Cat. #AM1906) according to manufacturer's instructions. The total RNA contained within each sample was quantified using the Qubit 4 Flurometer and the Qubit RNA BR (broad range) Assay Kit (Cat. #Q10210). The quality of each RNA sample was assessed using the Agilent 4200 TapeStation System and the RNA ScreenTape, RNA ScreenTape Ladder, and RNA ScreenTape Buffer (Cat. #5067–5576 to #5067–5578). Samples with a RIN higher than 8 were used for sequencing. Overall, 6 RNA samples per paternal or microinjection treatment group (n = 3 per sex) were sent for sequencing at AGRF. Library preparation was performed using the Illumina mRNA stranded protocol, and sequencing was conducted using the Illumina Novaseq 6000 SP workflow. 500 ng of RNA per sample was used for library preparation. All samples were run on a single flow cell lane using 150-bp-paired-end-reads.

**Hippocampus mRNA sequencing analysis.** The quality of reads was assessed using FastQC[67]. mRNA reads were trimmed against known Illumina adapter sequences using Trimmomatic[77]. Trimmed reads were then aligned to the mm39 reference genome using HISAT2 (v 2.2.1)[78]. An average of 96% of trimmed reads in each sample mapped to the GRCm39/mm39 reference genome. Aligned reads were summarized into counts using Rsubread's featureCounts[69]. Annotation of the RNAs was performed using the annotation file GRCm39 release M36 from GENCODE. Due to clustering of samples according to biological sex in multidimensional scaling plots in both F1 and microinjection

cohorts (Supplementary Fig. 6A, B), as well as sex-specific gene expression differences, we performed the male and female differential expression analyses separately. Multiple rounds of RNA extraction were performed across all F1 and RNA-microinjected samples and therefore "extraction round" was included as a batch effect in the differential expression analyses.

The DESeq2 package was used for the differential expression analyses between the paternal treatment groups[70]. Filtering of lowly expressed genes was performed using edgeR's 'Filter Genes by Expression Level' function[79]. The false discovery rate was controlled below 5% using the Benjamini and Hochberg method. A cut-off of $Log_2$ fold expression change of 1 and above (or −1 and below) relative to the control group was included in the analyses.

Functional annotation of the differentially expressed genes was conducted using Enrichr[40] to look at Kyoto Encyclopedia of Genes and Genomes (KEGG) pathways[72] and gene ontology pathways that were significantly regulated by these genes (FDR < 0.05). The figures were generated by the R package ggplot (v 3.4.4) and the pheatmap package (v 1.0.12).

**Statistical analyses**
General linear models were used for the normally distributed datasets in this study, with the exception of the bodyweight post-infection dataset in the male mice (Supplementary Fig. 1A) where a two-way ANOVA with Sidak's multiple comparisons test was used. The Shapiro-Wilk test of normality was used to evaluate the normality of each dataset and the residual plots were used to confirm normality and homoscedasticity. For F1/F2 natural mating cohorts, linear mixed models with 'sex' and 'paternal treatment' (or 'grand-paternal treatment' for F2) as fixed factors and 'litter' as a random factor were used for parametric datasets. Litter was incorporated as a random factor in natural mating studies since there may be decreased phenotypic variation between mice from the same litter. However, in cases where the random factor of litter contributed very little to the variance in the modeling of a particular outcome variable (< 10% of residual variance), a general linear model was used instead. General linear models were also used for the sperm RNA microinjection cohort. Main effects and interaction terms were assessed with the significance level alpha (α) = 0.05. *Post-hoc* pairwise comparisons were applied where significant interaction effects were present with a Bonferroni-Holm correction where applicable. A Mann-Whitney U-test was used for non-parametric data between two treatment groups. A Student's unpaired t-test was used for Shannon Index α-diversity. Cox regression analyses (proportional hazards) were applied for the novelty-suppressed feeding test with 'sex' and 'paternal SARS-CoV-2' as fixed covariates in natural mating studies, or 'sex' and 'microinjection received' as fixed covariates in the sperm RNA microinjection study. Statistical analyses were performed using R software (Rstudio version 4.2.2) with the use of packages tidyverse (v 2.0.0), readxl (v 1.4.2), lme4 (v 1.1–31), knitr (v 1.42), emmeans (v 1.8.5), parameters (v 0.20.2), performance (v 0.10.2), survival (v 3.4–0), survminer (v 0.4.9), coxme (v 2.2–18.1), and forcats (v 1.0.0). For RNA sequencing analysis, trimming and FastQC of FASTQ files were performed using Galaxy Australia (Australian Biocommons). All other RNA sequencing analysis steps were performed using R software (Rstudio version 4.2.2) with the use of packages stated above. Graphs were created using GraphPad Prism 8 (GraphPad software, LA Jolla, CA, USA) and R software (Rstudio version 4.2.2).

**Reporting summary**
Further information on research design is available in the Nature Portfolio Reporting Summary linked to this article.

## Data availability
The datasets generated in this study for Figures and the Supplementary Figs. are provided in the source data files. The RNA-sequencing

data (FASTQ files), including sperm small noncoding RNA and hippo-campus RNA sequencing datasets, generated in this study have been deposited in the NCBI Sequence Read Archive Database under accession code PRJNA1335862 for the sperm noncoding RNA dataset and accession code PRJNA1337639 for the hippocampus RNA dataset.

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

## Acknowledgements
We would like to thank Associate Professor Sue Finch from the University of Melbourne Statistical Consulting Platform for her guidance on the statistical methods used in this study. We would like to thank the WEHI Microinjection Service Co-ordinator Fiona Waters and her team for facilitating the fertilized oocyte microinjections and surrogate implantations. We would like to thank Brett Purcell for his help with repurposing behavioral assays for a PC3 facility. We would like to thank the Core Animal Services staff at the Florey Institute of Neuroscience and Mental Health for their help with husbandry. We would like to thank Dr Stefanie Bader, Dr James Cooney, Reet Bhandari, Le Wang, and Jan Schafer for their help with animal welfare checks within the PC3 facility. F.Z.M. is supported by a Senior Medical Research Fellowship from the Sylvia and Charles Viertel Charitable Foundation, a National Heart Foundation Future Leader Fellowship (105663), and National Health and Medical Research Council (NHMRC) Emerging Leader Fellowship (GNT2017382). S.N.R is supported by an Erwin Schrödinger Fellowship (J4776-B) from the Austrian Science Fund (FWF). This research was funded by an NHMRC Ideas Grant to A.J.H., who has also been supported by an NHMRC Principal Research Fellowship, and a DHB Foundation (Equity Trustees) grant to A.J.H.

## Author contributions
Elizabeth A. Kleeman (E.A.K.) performed and analyzed all behavioral and physiological experiments in the F1/F2 natural mating cohorts. E.A.K. also performed breeding and general animal husbandry within the PC3 facility, E.A.K. repurposed all behavioral tests for the PC3 facility. E.A.K. performed all the work and analyses related to the F1 hippocampal transcriptomics, F1 immune challenge, and sperm small noncoding RNA analyses. E.A.K. also prepared RNA solutions and performed most of the behavioral tests in the RNA-microinjection study (including all subsequent analyses). E.A.K. participated in all tissue dissections. E.A.K. also wrote the first draft of this manuscript. Dr Carolina Gubert (C.G.) played a major role in study conceptualization, experimental study design across all studies, student supervision and supported data analysis. C.G. also participated in tissue dissection for the microinjection studies. C.G. played a major role in proofreading this manuscript. Dr Sonali N Reisinger (S.N.R.) also played a major role in study conceptualization, experimental study design across all studies, and student supervision. S.N.R. also participated in tissue dissection and measured brain weight for the microinjection studies. S.N.R extracted RNA for microinjection studies. S.N.R. played a major role in proofreading this manuscript. Dr Kathryn Davidson (K.D.) managed and coordinated activities within the PC3 facility and participated in protocol design for all experiments conducted within PC3. K.D. also proofread this manuscript. Da Lu (D.L.) helped analyse sperm small noncoding RNA datasets and performed the piRNA cluster analysis. D.L. also proofread this manuscript. Merle Dayton (M.D.) and Liana Mackiewicz (L.M.) performed experiments related to the infection of the male mice, assisted with breeding for F1/F2 cohorts, general animal husbandry, and tissue dissection within the PC3 facility. M.D. and L.M. also proofread this manuscript. Bethany A. Masson (B.A.M.) performed and helped analyse all cognitive experiments in the microinjection study. B.A.M. also proofread this manuscript. Pranav Adithya (P.A.) assisted in performing some behavioral video analysis for the natural F1/F2 mating studies. P.A. also proofread this manuscript. Dr Alexandra Garnham (A.G.) assisted with the bioinformatics analysis for the sperm small noncoding RNA sequencing datasets. A.G. also proofread this manuscript. Dr Gemma Stathatos helped with experiments related to the testes histological examination and also proofread this manuscript. Professor Moira O'Bryan helped with experiments related to the testes histological examination and also proofread this manuscript.

Rikeish R. Muralitharan analysed maternal microbiome datasets and also proofread this manuscript. Francine Z. Marques helped analyse maternal microbiome datasets and also proofread this manuscript. Shanshan Li (S.L.) assisted with the small RNA enrichment protocol design for the microinjection study. S.L. also proofread this manuscript. Shae McLaughlin (S.M.), Emmet T Keough (E.T.K), and Michelle Wheeler (M.W.) assisted in performing some behavioral video analysis for the natural F1/F2 mating studies. S.M., E.T.K and M.W. also proofread this manuscript. Pamudika Kiridena (P.K.) assisted with tissue dissection in the microinjection study and study conceptualization. P.K. also proofread this manuscript. Dr Marcel Doerflinger (M.D.) is a laboratory head who participated in conceptualization and providing resources, including laboratory space, instrumentation and reagents, to support research activities within the natural F1/F2 experimental cohorts. M.D. also proofread this manuscript. Professor Marc Pellegrini (M.P.) is a former laboratory head who participated in conceptualization, obtaining funding for this project, and providing resources including laboratory space, instrumentation and reagents, to support research activities. M.P. also proofread this manuscript. Professor Anthony J. Hannan (A.J.H.) is the chief investigator for this project. A.J.H. was responsible for obtaining funding for the project, providing laboratory resources, study conceptualization, project management, experimental design, data interpretation and supervision. A.J.H. also played a major role in editing and proofreading this manuscript.

## Competing interests
The authors declare no competing interests.

## Additional information

Elizabeth A. Kleeman[1,2], Carolina Gubert [1,10] ✉, Sonali N. Reisinger [1,10], Kathryn C. Davidson[3,4], Da Lu[1,2], Merle Dayton[3], Liana Mackiewicz [3], Bethany A. Masson[1,2], Pranav Adithya[1,2], Alexandra L. Garnham[3,4], Gemma Stathatos [5], Moira K. O'Bryan [5], Rikeish R. Muralitharan [6,7], Francine Z. Marques [6,7,8], Shanshan Li[1], Huan Liao[1], Shae McLaughlin [1], Emmet T. Keough[1], Michelle Y. Wheeler[1], Pamudika Kiridena[1,2], Marcel Doerflinger [3,4], Marc Pellegrini [3,4] & Anthony J. Hannan [1,2,9] ✉

[1]Florey Institute of Neuroscience and Mental Health, Parkville, Victoria, Australia. [2]Florey Department of Neuroscience and Mental Health, University of Melbourne, Parkville, Victoria, Australia. [3]The Walter and Eliza Hall Institute of Medical Research, Parkville, Victoria, Australia. [4]Department of Medical Biology, University of Melbourne, Parkville, Victoria, Australia. [5]The School of BioSciences and Bio21 Molecular Science and Biotechnology Institute, University of Melbourne, Parkville, Victoria, Australia. [6]Hypertension Research Laboratory, Department of Pharmacology, Biomedical Discovery Institute, Faculty of Medicine, Nursing and Health Sciences, Monash University, Victoria, Australia. [7]Victorian Heart Institute, Monash University, Melbourne, Australia. [8]Baker Heart and Diabetes Institute, Victoria, Australia. [9]Department of Anatomy and Physiology, University of Melbourne, Parkville, Victoria, Australia. [10]These authors contributed equally: Carolina Gubert, Sonali N. Reisinger. ✉e-mail: carolina.gubert@florey.edu.au; anthony.hannan@florey.edu.au

