## [Transparent Peer Review file · Nature Communications]

Paternal SARS-CoV-2 infection impacts sperm small noncoding RNAs and increases anxiety in offspring in a sex-dependent manner

Corresponding Author: Professor Anthony Hannan

Version 1:

Reviewer comments:

Reviewer #1

(Remarks to the Author)

The current study by Kleeman et al investigates the effects of paternal SARS-CoV-2 infection on offspring in a rodent model. They assessed sperm RNA payload in fathers and behavior and hippocampal gene expression in the offspring.

Noteworthy results/Significance to the field/comparison to established literature:

The novelty of this study lies within the choice of exposure. No studies on intergenerational effects so far have assessed the impact of Covid 19 infection. On the contrary route of transmission (RNA mediated germline inheritance), assessed carrier molecules (miRNAs and piRNAs authors themselves, Rodgers et al, Qi Chen et al, Rando et al Gapp et al), affected phenotypes (anxiety-like behavior – the authors themselves and many others including Rodgers et al, Qi Chen et al, Gapp et al.,) and system affected by the exposure (immune system – the authors themselves and Hackett et al) have been studied in the context of other exposures before.

General comments/suggestions:

1. The authors nicely introduce the very distinct manifestations of Covid 19, with those individuals experiencing severe neurological conditions such as severe brain fog or debilitating pain, representing a smaller percentage. I am wondering why the current study mentions those extremes, that are unlikely to sire offspring in real life and that the current mouse model doesn't capture since the animals seem to recover well.
2. For less severely affected cases - given the strong psychological burden that several imposed isolation measures have caused I am wondering about the effect sizes relative to the observed results.

Writing:

The text is extremely well written, has a nice flow and follows a very clear logic. It was a pleasure to read it!

Minor point:

179: to enter the light/dark box – I assume should say the light compartment

Figures:

- Great that single data points are depicted

Page 56: J refers to something that should have 4 colors, but J only has 2 colors

Results:

- The infection and their inclusion criteria mention a strong reduction in body weight, which in itself is a strong inducer of intergenerational effects (work by Josep Jimenez Chillaron in Barcelona who uses artificial control of litter sizes to induce the weight differences) – to me it is unclear how thus other types of infection can be disentangled from those weight effects. To me this is a severe methodological flaw for which I currently see no remedy. Even if the bodyweight is recovered 8 days post infection and breeding takes place 4 weeks post infection the germ cells must have been affected by the body weight difference.

- For the amount of different behavioral tests assessed there are very few significant effects with small effect size and besides one case, not very strong p-values. These can of course still reflect important effects, if they withstood multiple testing correction. Have these effects been withstanding multiple testing correction?

- mRNA sequencing of the offspring brain– to justify the male/female separate analysis it would be important not only to visualize the different samples in a PCA (as done) but also conduct an overall analysis – that would also give the authors more power to detect interactions between sexes for each gene product. The methods also mention surrogate variables to control for potential batch effects. What would those batches be?

- Sperm small RNA seq: only 22 small RNAs are affected - this is a very small number across all small RNA classes. I mostly wonder about the piRNA analysis – in my experience it is not meaningful to analyze single piRNAs – yet they should show alterations at the cluster level. Have the authors analyzed the piRNAs as clusters? I cannot imagine a modus operandi for single piRNAs and if such is eluded to it should be substantiated. Also the small RNA sequencing usually doesn't assess the size typical of piRNAs during the library preparation size selection step. The miRNAs could be easily validated by q-PCR?

- Study design: Is it possible to rule out routes of transmission via mating behavior, mother/father carry over of microbiota etc, other immunological remnants of infection (antibodies, Cort levels in the seminal fluid see Teperino lab studies, or studies by Adam Watson's lab)

- How many litters were used –has this effect of transmission been replicated? Is each offspring n referring to a litter or to a litter mean or to a single offspring animal (is each animal coming from a different litter?). It is currently unclear from the methods section how different timings required to reach peak levels of viral load were accounted for in the mating scheme? Parental age could also be specified further – it currently states: "at least.....". Were the animals age matched?

- Microinjection experiment: To me it seems the outcome doesn't partially replicate (one single readout of one behavioral test is mimicking the natural offspring). At most this indicates that the isolated sperm fractions could be functional and should be stated differently. The small RNA enriched sample shows substantial amounts of intact ribosomal RNA that likely indicate somatic cell contamination. It would be important to see all traces prior enrichment and/or other measures of sperm purity to exclude contamination.

Comment on sex and gender: The title does currently not reflect that some aspects of the phenotype are sex specific. Currently no source data provided

The data availability statement is not conform with nat. comm guidelines. If I as a reviewer would want to assess the data I'd first have to make a "reasonable request".

Reviewer #2

(Remarks to the Author)

The manuscript by Kleeman et al. investigates the very relevant and timely topic of the effects of pre-conceptual exposure to SARS-CoV-2 and its consequences on anxiety-like behavior in mice. In general, the study is well-designed and organized, and I appreciate the substantial amount of work and planning involved in transgenerational studies. However, I have several concerns about the results and data interpretation, as I believe the general conclusions of the study are bold based on the results presented in this manuscript.

I am wondering why the authors decided to report only testicular weight. It would be useful for the study to include some sperm characteristics of infected and non-infected males or to make histological sections of the testes to show if there are differences in the morphological appearance of cells at different stages of spermatogenesis. Additionally, body weight data of the F1 generation should be presented based on sex, as the offspring's sex can be easily determined by anogenital distance from PND 8.

I am also curious why the authors measured the whole brain weight rather than evaluating a volume estimation of the hippocampus. This would be a more appropriate readout, as changes in hippocampal volume may reflect structural plasticity that could be associated with behavioral changes. A better approach might be for the authors to dissect the hippocampus bilaterally and then randomly use hippocampi from one side of the brain for RNA sequencing and the other for histology.

Regarding the RNA-seq data, it is surprising that only a small number of differentially expressed genes were found. This could be due to the bulk analysis performed (I understand that single-cell analyses are expensive), but it might be better if the authors attempted to dissect the ventral and dorsal hippocampus separately before running the bulk analysis, as these regions can have distinct gene expression profiles.

Any type of enrichment analysis is not relevant with such a small number of differentially expressed genes, and the statement about an altered hippocampal transcriptome in females is too bold, which such a low number of differentially expressed genes. The most interesting part of the study is actually the sperm analysis, even though some key sperm

characteristics like morphology, viability, and motility are missing. This section could be presented even before the hippocampal or behavioral data, as it is a crucial part that confirms sperm is affected by SARS-CoV-2 infection. However, due to individual variation between even genetically identical mice, the downside is the pooling of mouse samples for sequencing. Also, the behavioral phenotype after the microinjection experiment is a bit more convincing (but again, there is a question about correction for multiple testing). It would also be interesting to see if, after the microinjection experiment, the situation in the hippocampus would remain the same. If the same genes are expressed (even if it's still a small number of genes), that would truly confirm that the results are a consequence of the treatment.

Therefore, even though this is a truly novel and important experiment, I would advise the authors to reanalyze the behavioral testing, correct for multiple testing, or use behavioral score, and rewrite the bold statement about the link between infection and anxiety-like phenotypes. Again, as a strong supporter of publishing null findings, since they are equally important, it would be good for the authors to reanalyze the results, rather than focus on finding a couple of genes and making bold statements.

Reviewer #3

(Remarks to the Author)

It is an interesting paper, and it will have a potential impact on the field. However, some improvements need to be made.

1. It is hard to interpret Figures 1, 3, and 5. For example, in Figure 1, considering that the only significant values indicated by (*) on the top graph are the values for the male group in Figure 1D and Figure 1L. The post hoc analysis p-values are shown below the graph. I suggest that adding the p-values of significant post hoc analyses to the top of the graph will help to avoid confusion in the interpretation.
2. Figure 1L presented in the main figure is not necessary here, it is better to move to the Supplementary figure.
3. Figure 1O is confusing, the authors say that there is no difference in immune response but on the figure, there are significant changes between control and treatment, so it is better to compare the increased ratios Poly I:C compared to saline in each group.
4. Surprisingly, the authors determined very few genes that were differentially expressed in F1 females, and no differentially expressed genes were found in males. Based on Figures 1A and 1L, males showed more alterations in behavior. How could the authors explain that?
5. I am also not convinced by the mating time after infection. The full period of spermatogenesis in mice is 7 weeks. Why do the authors mate the mice after 4 weeks after the infection?
6. How was the hippocampus isolated from the whole brain, from fresh or frozen tissue? The precision of dissection could affect the RNA-seq analysis and impact the determination of differentially expressed genes.
7. Which quantity of the RNA was taken for RNA-seq analysis?
8. Which Fold Change (FC) cut-off was used for RNA-seq results in both RNA-seq and small RNA sequencing experiments

Version 2:

Reviewer comments:

Reviewer #1

(Remarks to the Author)

In the revised manuscript titled "Paternal SARS-CoV-2 infection impacts sperm small noncoding RNAs and increases anxiety in offspring in a sex-dependent manner," the authors Kleeman et al. have made an effort to address all the issues raised. While I find the revised manuscript improved, there are some remaining concerns about the robustness that should be addressed so that I could recommend publication without reservation.

The authors have added important additional clarifications in the text. They have further analyzed weight trajectories in the offspring. Behavior was reanalyzed to summarize individual tests by converting measures into z-scores. The z-score confirmed modest alterations in anxiety behavior (very significant effect, $p=0.003$) and a sex-specific z-score summary of depressive-like behavior. This is interesting given the molecular sex-dependence of hippocampal gene expression. Yet caution is warranted in the interpretation thereof, since the individual test of depressive-like behavior did not show those effects. With the z-score, no power is gained; thus, the authors should explain why the results obtained by computing the z-score in this case doesn't reflect a finding by chance, especially since the F2 offspring shows a similar outcome.

Overall, even though the authors reduced the number of multiple comparisons by computing z-scores, the issue remains that out of the behavioral phenotypes assessed (light-dark box test, open field test, novelty suppressed feeding, sucrose consumption, social interaction, novel-object recognition, y-maze, summarized into: anxiety, cognition, social interaction, depressive-like, and feeding behavior), only one showed a convincingly significant change (the anxiety behavior). So there is a 20.36% chance of getting at least one false positive; in other words, had this experiment been done five times, once it would show the effect by chance. In that sense, the RNA-injection finding on the anxiety behavior can somewhat strengthen the argument of a robust effect, yet it seems to be sex-dependent. So I am not fully convinced that we are looking at a replicable change. It might be more relevant to see whether those effects can replicate if repeated, rather than relying on the provided z-score analysis, to gauge whether those effects do not reflect a false positive. In the rebuttal, the authors mention that the phenotype has been observed in multiple cohorts, yet the data of both cohorts have been pooled. Seeing the

replication in a distinct cohort would make a very strong case and be very convincing. To this end, it would be necessary to see the analysis of both cohorts separately. Also, could the absence of other significant measures not be due to different power (if in anxiety we are looking at 16-22 animals and in the other tests 10-12)?

How do the authors explain the change in fasting-induced weight loss? Did they check energy expenditure or glucose/insulin tolerance? I do not consider this very crucial - but would be interested to know.

The authors made a great addition to their sperm RNA analysis by analyzing their piRNAs at the cluster level. They furthermore predicted putative targets of the single tRNA fragment and miRNA found to be differentially abundant in mature sperm. Can they replicate the miRNA finding by independent sequencing of sperm or by q-PCR in an independent cohort?

The authors relate the behavioral and molecular sex-dependent findings. Support of this conclusion would require seeing that some of the gene expression changes are indeed involved in the anxiogenic phenotype following the same pattern.

It is great that the authors conducted another round of gene expression analysis – how did it compare to the first round? Even if mapping rates were worse before, were some of the prior findings replicable? Thanks for having clarified what was referred to by batch effects.

It is interesting and fitting that some differentially regulated genes are known to play a role in anxiety-related behaviors. If there is a sex-independent behavioral effect at the z-score level related to hippocampal gene expression and those genes are indeed involved in the present model, why would they not be differentially regulated in the males as well? Do they show the same tendency? Are they known to be sex-dependently regulated? If the gene expression is analyzed for an interaction between sex and treatment, no significant genes are revealed – to me that means that no independent analysis separated by sex is indicated. Should the differentially regulated genes be at all related to the observed anxiety behavior, why does the RNA injection not recapitulate more of those changes?

In terms of the rebuttal response to the point of why extreme cases should or should not be mentioned in the introduction, the authors argue that the model does produce those cases. That is not the point of my criticism. The point is: is the present study using those severe cases? If not, why are they brought into the picture? Even if the mice used in the study are simply too young to see those severe phenotypes, then those severe phenotypes are not the subject of the current study and should not be emphasized. But this is not an essential point to me - I simply find it misleading.

For less affected cases as studied in the manuscript: even if the sperm RNA profile is distinct from the one produced by stress alone – in this model, the sperm RNA profile does not seem to be a strong driver of intergenerational change in the main anxiety z-score, nor are the sperm RNA changes particularly strong (with piRNA changes appearing to show the strongest effects, yet their involvement in mammalian transmission is yet to be proven in the field). Samples 14, 15, and 18 don't look consistent with the other samples in that they show considerable ribosomal RNA peaks – as high as the small RNA fraction. Do the authors have orthogonal measures of purity? What samples do those "outliers" correspond to in the heatmap Fig.2. The pasted bioanalyzer plots were too blurry to read this accurately.

I appreciate the authors recognizing the limitation with regards to the distinction between body weight-induced effects and infection-induced effects. I wonder though whether comparing the sperm RNA levels here is meaningful again due to their apparently limited involvement in the transmission even if functional. At the same time, to my understanding, the authors did not assess metabolic alterations in the current model in the offspring which could be present, and as the authors pointed out, their own prior model also observed behavioral effects in the offspring, even though a comparison of models seems hard, since maternal behavior was affected in the nutritional model. Currently, to me, it remains hard to conclude from the present data that the infection here is different from body weight-induced changes. Pointing this out as a limitation in writing is important, yet I am not convinced that is sufficient. Can the authors try to adjust offspring behavioral measures with the paternal weight as a covariate? Showing a persistent effect would make a strong case.

It is great that the authors could rule out SARS-CoV-2 transmission itself on the females and affected microbiota! Thanks for clarifying the study design with regards to statistical units, age, etc. This appears clear now.

Reviewer #2

(Remarks to the Author)

The authors have been very responsive to my comments and thanks a lot for that. Thanks to the all comments from others and the authors' responsiveness, the manuscript is now in great shape and I am very much looking forward to seeing this important and interesting work officially published soon.

Reviewer #3

(Remarks to the Author)

The authors addressed the raised questions accordingly.

Version 3:

Reviewer comments:

Reviewer #1

(Remarks to the Author)

The authors have addressed all my concerns in detail. Thank you.

Response to Editor comments

We thank the Associate Editor, Dr Brittany Davis, for her highly constructive comments and helpful feedback on our submitted manuscript. We have now revised our manuscript to address the comments from Dr Davis, as detailed below.

EDITOR'S COMMENTS:

Topically, mechanisms of intergenerational epigenetic transfer are of high interest for the journal and a personal interest of mine. That is to say, we are generally very interested in this type of work. However, this area of research is still quite underdeveloped and making strong claims requires very strong evidence, which you have not provided.

You consistently use the term epigenetic mechanisms throughout, but the identification of non-coding RNA expression changes does not constitute an epigenetic change. It may lead to or be a consequence of epigenetic modifications. To publish at Nature Communications with such claims we would really need more of the mechanism here.

One path forward if you do not plan to add more data in the current version is to reduce the claims/consistent ref to epigenetics unless you show which modifications/mechanism; and limit overinterpretations.

RESPONSE:

I greatly appreciate these insightful and constructive comments, and your interest in our work. When planning to publish such novel findings, with major public health implications, it is always a delicate balance between the need to urgently communicate the important findings with the international community, and the valuable resources and time required to generate additional data prior to publication, as well as the career impacts on the postdocs and graduate students involved.

We have taken into account all of the advice you have provided, including rewording the text with respect to epigenetic inheritance. We have also avoided potential overinterpretation, as you have so kindly suggested.

In revising the attached manuscript we have incorporated the very helpful feedback and advice you have provided, including altered language associated with interpretation of the effects of altered sperm RNAs on offspring phenotypes. Furthermore, we have revised the text so as to limit overinterpretation, in response to your constructive comments. We have also modified the manuscript to only refer to epigenetics in its strictest definition. We feel that the further text changes will avoid any perceived overinterpretation of the data in the manuscript.

We strongly feel that the findings in our manuscript do constitute a striking advance, with wide implications and major potential impacts that meet, or exceed, those of many research articles published in your journal in recent years. Furthermore, due to the novelty and significance of our findings, we feel that it would attract a broad readership and substantial citations.

The finding that SARS-CoV-2 infection and COVID-19 illness specifically alter information in sperm (small RNAs) that transfer information to the next generation, tightly coupled to striking phenotypic changes in the offspring (including increased anxiety), represents a major advance that will have impacts across multiple areas of biology and medicine. The fact that our manuscript also includes a key mechanistic experiment, where small RNAs were microinjected into fertilized oocytes, and the resultant offspring recapitulated aspects of the epigenetic inheritance observed following natural mating, greatly increases the importance of the findings, including their translatability to humans (where highly similar small RNAs are also found in sperm, and can be modulated by environmental exposures).

The discoveries communicated in our manuscript will inform future studies aimed at establishing the global impacts of the COVID-19 pandemic on the next generation, including relevance to anxiety disorders and other aspects of mental health. Our findings also have wider implications for paternally mediated epigenetic inheritance following other viral infections and associated illnesses. This could lead to novel interventions associated with future global epidemics and pandemics.

In summary, we strongly feel that our manuscript deserves to receive peer review at your journal, and those expert assessments from international reviewers would help you better assess the novelty, significance and impact of our new findings.

Response to Reviewers

We thank the Editor, and all of the Reviewers, for their highly constructive comments to further improve our revised (R1) manuscript. In the attached further revised manuscript we have fully addressed each and every one of the latest comments from each Reviewer regarding our revised manuscript. Below we describe how we have described in detail how we have fully addressed each of the latest Reviewer comments.

Reviewer #1

The current study by Kleeman et al investigates the effects of paternal SARS-CoV-2 infection on offspring in a rodent model. They assessed sperm RNA payload in fathers and behavior and hippocampal gene expression in the offspring.

Noteworthy results/Significance to the field/comparison to established literature:

The novelty of this study lies within the choice of exposure. No studies on intergenerational effects so far have assessed the impact of Covid 19 infection. On the contrary route of transmission (RNA mediated germline inheritance), assessed carrier molecules (miRNAs and piRNAs authors themselves, Rodgers et al, Qi Chen et al, Rando et al Gapp et al), affected phenotypes (anxiety-like behavior – the authors themselves and many others including Rodgers et al, Qi Chen et al, Gapp et al.,) and system affected by the exposure (immune system – the authors themselves and Hackett et al) have been studied in the context of other exposures before.

General comments/suggestions:

The authors nicely introduce the very distinct manifestations of Covid 19, with those individuals experiencing severe neurological conditions such as severe brain fog or debilitating pain, representing a smaller percentage. I am wondering why the current study mentions those extremes, that are unlikely to sire offspring in real life and that the current mouse model doesn't capture since the animals seem to recover well.

RESPONSE: Although most mice recover from the P21 SARS-CoV-2 strain in terms of their bodyweight, the P21 model does show signs of severity in other aspects. This mouse model is described extensively in Bader et al. 2023 where they show that the P21 model exhibits cytokine storm in the lungs (panel D below), severe vasculitis, perivascular inflammation, inflammation of alveolar ducts, and multifocal inflammatory infiltrates (neutrophils, macrophages, and T cells) in the alveoli, vessels, and subpleural space even after their bodyweight has recovered (panel C below)(Bader et al. 2023, *PNAS*, PMID: 37523564). This model also shows lethality in older mice (panel A and B below) (aged 6-8 months), which is consistent with a moderate-to-severe infection of SARS-CoV-2 in humans. The authors of Bader et al. (2023) also have another manuscript published at *Nature Communications* demonstrating that this mouse model shows signs and symptoms of long COVID or post-

acute sequelae of COVID-19 (Bader et al. 2025, *Nature Commun.*, PMID: 40180914). Therefore, it is appropriate that we mention those extremes as they are consistent with the P21 model used here. Also, it is still quite reasonable to assume that those who have experienced a moderate SARS-CoV-2 infection are still able to reproduce after some time has lapsed from infection since we saw no obvious signs of reduced functional fertility in the mice after 4 weeks post-infection.

A) Survival curve for aged P21 mice, **B)** Weight loss trajectory in P21 model with young and aged mice, **C)** Lung H&E staining of the lungs of P21 infected mice showing damage, **D)** cytokine storm in the lungs of P21 mice. Taken from Bader et al. 2023.

For less severely affected cases - given the strong psychological burden that several imposed isolation measures have caused I am wondering about the effect sizes relative to the observed results.

RESPONSE: As this Reviewer points out, it can be difficult in humans to dissociate the intergenerational effects of stress of the pandemic from the effects of the infection itself. Furthermore, having a moderate to severe viral infection is usually quite stressful. Nonetheless it is still valuable to investigate the effects of SARS-CoV-2 and other infections on the sperm RNA code and offspring phenotypes, particularly since we see changes in the sperm RNA code that do not overlap with other paternal stress models (Short et al. 2016, *Trans Psych*, PMID: 27300263, Gapp et al. 2014, *Nat Neurosci*, PMID: 24728267) . Furthermore, the growing paternal infection studies (Kleeman et al. 2024, *Brain Behav Immun*, PMID: 37820975, Tyebji et al. 2020, *Cell Rep*, PMID: 32348768, and Liao et al. 2024, *Brain Behav Immun*, PMID: 38636562) reveal that there are distinct effects of each paternal infection on sperm small RNAs and offspring behavioural phenotypes. In fact, there are no overlapping phenotypes or sperm RNA changes between the paternal Poly I:C and paternal SARS-CoV-2 models (Kleeman et al. 2024). If the stress associated with infection/inflammation or bodyweight change were the main drivers in these paternal models, we would expect to see more overlap between these different infection paradigms.

Furthermore, we see relatively robust effect sizes across key parameters in the F1 cohort as shown in the table below. Also, we have included behavioral z-scores for each behavioral domain we have investigated. The anxiety z-score in the F1 offspring of SARS-CoV-2 infected male mice is particularly robust ($(F_{(1, 71)} = 9.166, P=0.003, \text{Figure 1F})$).

Parameter	Effect size (Cohen's d)	Males only in I	Interpretation
Change_in_BW_post_fast	0.992341357	1.550004717	Large
time_spent_centre	-0.631504257		Medium
time_in_light_zone	-0.521792511		Medium
Latency_light	0.303245005	0.611528081	Medium

Writing:

The text is extremely well written, has a nice flow and follows a very clear logic. It was a pleasure to read it!

Minor point:

179: to enter the light/dark box – I assume should say the light compartment

RESPONSE: This has been corrected.

Figures:

Great that single data points are depicted

Page 56: J refers to something that should have 4 colors, but J only has 2 colors

RESPONSE: All legends have been corrected to address this.

Results:

The infection and their inclusion criteria mention a strong reduction in body weight, which in itself is a strong inducer of intergenerational effects (work by Josep Jimenez Chillaron in Barcelona who uses artificial control of litter sizes to induce the weight differences) – to me it is unclear how thus other types of infection can be disentangled from those weight effects. To me this is a severe methodological flaw for which I currently see no remedy. Even if the bodyweight is recovered 8 days post infection and

breeding takes place 4 weeks post infection the germ cells must have been affected by the body weight difference.

RESPONSE: The Reviewer does make a good point here and we have considered this. With regards to work by Josep Jimenez Chillaron, they did detect metabolic changes in the offspring and grand-offspring of fathers that experienced neonatal overgrowth, suggesting that paternal overnourishment can have transgenerational consequences (Jimenez-Chillaron et al. 2016, *Proc Nutr Soc*, PMID: 26573376). Our lab has also shown that paternal western diet can have intergenerational behavioural effects too (Bodden et al. 2022, *FASEB J*, PMID: 34907601). Therefore, body weight and metabolic changes can influence paternal epigenetic inheritance and we have pointed this out as a limitation in our model and other infection models (see discussion section). Nonetheless, when we compare our results with other paternal infection/inflammation and paternal diet/weight change models (Liao et al. 2024, Bodden et al. 2022, Carone et al. 2010, *Cell*, PMID: 3039484), we see very distinct offspring phenotypes and changes to the sperm RNA code. For example, in the paternal lipopolysaccharide (LPS) exposure model published recently (Liao et al. 2024), the LPS exposed F0 male breeders lost a similar amount of weight (10-15%) from the inflammation when compared to our SARS-CoV-2 infection model during the acute inflammatory stage. Yet we see almost completely distinct behavioural phenotypes, since they reported decreased anxiety in their F1 female mice from LPS exposed fathers. Also, there is very little overlap in the sperm small RNA profiles between each model. Although not conclusive, this seems to suggest that different types of immune activation rather than bodyweight loss alone are leading to distinct F1 phenotypes and associated sperm RNA changes.

For the amount of different behavioral tests assessed there are very few significant effects with small effect size and besides one case, not very strong p-values. These can of course still reflect important effects, if they withstood multiple testing correction. Have these effects been withstanding multiple testing correction?

RESPONSE: We have calculated effect sizes (Cohen's d) for each behavioral measure that is significant in the F1 offspring and they range from medium to large in size. This indicates that they are robust:

Parameter	Effect size (Cohen's d)	Males only in I	Interpretation
Change_in_BW_post_fast	0.992341357	1.550004717	Large
time_spent_centre	-0.631504257		Medium
time_in_light_zone	-0.521792511		Medium
Latency_light	0.303245005	0.611528081	Medium

RESPONSE: There are cases where multiple test corrections may be prudent, such as when a large number of endpoints are tested within a single behavioural test (Benjamini et al. 2001, *Behav Brain Res.*, PMID: 11682119). In our case, we have only a few endpoints (in most cases only one endpoint) for each behavioural test in our battery of six tests.

Nonetheless, the calculation of Z-scores across multiple tests related to the same behavioural domain is a valuable and evidence-based way to take into account behavioral profiles across multiple tests and endpoints (Guilloux et al. 2011, *J Neurosci Methods*, PMID: 21277897). To apply this more stringent and convincing behavioural analysis, we have computed behaviour Z scores for anxiety based on a previously published protocol (Guilloux et al. 2011). Results from each outcome measure are normalised to generate scores for each test and combined to give each mouse a single unified score for each behavioral trait. In our case, we have computed scores for anxiety, sociability, cognition, feeding behavior, and depression. As you can see, anxiety scores were significantly higher for offspring of SARS-CoV-2 infected fathers. These results have been integrated into the revised manuscript.

Overall, we believe our heightened F1 anxiety result to be robust as we detected multiple F1 anxiety traits across several tests (light/dark box and open field test) and endpoints within the light/dark box (both time spent in light zone and latency to light zone). Also, these anxiety measures have been tested across multiple cohorts as mentioned in the methods section under “F1/F2 experimental timelines”. Furthermore, we have also recapitulated the latency to enter light zone result in the RNA microinjection cohorts. The behavior z-scores also reveal that anxiety has been altered by paternal SARS-CoV-2 infection ($F_{(1, 71)} = 9.166$, $P = 0.003$, **Figure 1F**), with a strong p-value. We have also been conservative and stringent with our statistical analyses by incorporating litter effects in the linear mixed models to ensure that the significant effects we see are robust. It is therefore highly unlikely that these significant results are false positives.

mRNA sequencing of the offspring brain– to justify the male/female separate analysis it would be important not only to visualize the different samples in a PCA (as done) but also conduct an overall analysis – that would also give the authors more power to detect interactions between sexes for each gene product. The methods also mention surrogate variables to control for potential batch effects. What would those batches be?

RESPONSE: We updated improved our analyses by conducting paired end RNA-sequencing which resulted in higher mapping rates (~97% of reads mapped to GRCm39/mm39 genome) and annotation rates (~65-70% reads were annotated by GRCm39/m39 gencode M35 primary assembly) compared to our previous single end RNA-sequencing analysis. As suggested, we conducted an overall analysis using DeSeq2 and saw no significant interactions between sex and paternal treatment for any gene in both the F1 and RNA-microinjected analyses.

Interaction model F1 analysis (top genes):

	baseMean	log2FoldChange	pvalue	padj
Adgrb2	10493.79921	0.5598033	0.0000054	0.1008390
Gse1	570.87817	0.8607983	0.0001262	0.9695723
Gm13302	16.08362	3.1509728	0.0001560	0.9695723
Gnai3	1261.37589	-0.0038519	0.9822645	0.9999604
Cdc45	44.61996	0.2362552	0.5929513	0.9999604

Interaction model RNA microinjected analysis (top genes):

	baseMean	log2FoldChange	pvalue	padj
1700048020Rik	152.43819	-1.3677155	0.0000835	0.7918876
Scn4b	1053.01018	-0.4374633	0.0000889	0.7918876
Gnai3	1222.59301	-0.0477181	0.6618213	0.9999596
Cdc45	58.89601	-0.4094026	0.2854551	0.9999596
Scm12	37.66854	-0.0435059	0.9406233	0.9999596

However, there were a number of significantly different genes between the sexes (see below). This led us to separating the sexes, which is a valid approach used in the literature (Fiorini et al. 2022, *Front Neurol*, PMID: 36619915, Rai et al. 2022, *J Orthop Res*, PMID: 35266580).

Male vs. female F1 analysis (top genes):

	baseMean	log2FoldChange	pvalue	padj
Kdm5d	571.53720	10.1759869	0.0000000	0.0000000
Xist	12058.09976	-11.0239013	0.0000000	0.0000000
Gm29650	100.74329	7.6281637	0.0000000	0.0000000
Uty	297.36069	8.7105464	0.0000000	0.0000000
Ddx3y	1195.93721	9.5610865	0.0000000	0.0000000
Flrt2	2195.87504	0.4246656	0.0000000	0.0000001
Creb5	248.65922	0.7398124	0.0000000	0.0000006
Foxo6	438.09971	-0.7451595	0.0000001	0.0001468
Lrfn2	562.72747	-0.3750560	0.0000001	0.0001778
Dipk2a	1258.26312	0.5067691	0.0000012	0.0020758
Gm42756	108.28520	0.6655479	0.0000031	0.0047167
Hs3st1	440.20015	0.5360641	0.0000062	0.0087101
Gmip	259.71164	-0.4511224	0.0000094	0.0122334
Pde7b	697.94068	0.4844874	0.0000140	0.0169290
Sst	5626.37899	-0.3014276	0.0000167	0.0187884
Kctd6	1686.21415	-0.2864963	0.0000260	0.0273984
Eif2s3x	1502.87778	-0.4071706	0.0000429	0.0426460
9330111N05Rik	26.23761	1.2076581	0.0000490	0.0459462

Male vs. female RNA-microinjected analysis (top genes):

	baseMean	log2FoldChange	pvalue	padj
Xist	16898.4429	-12.5872708	0.0000000	0.0000000
Ddx3y	1007.0214	13.5851011	0.0000000	0.0000000
Eif2s3y	738.3683	13.1392041	0.0000000	0.0000000
Kdm5d	529.6097	12.6593199	0.0000000	0.0000000
Kdm5c	2735.0877	-0.4095612	0.0000000	0.0000000
Kdm6a	1381.6200	-0.5232376	0.0000000	0.0000000
Eif2s3x	2326.6707	-0.5522037	0.0000000	0.0000000
Ddx3x	11181.2494	-0.3069011	0.0000000	0.0000000
Rab10os	1281.3079	-0.4040979	0.0000000	0.0000000
5530601H04Rik	569.0563	-0.5615382	0.0000000	0.0000001
Lrp1	22349.4668	-0.2227606	0.0000001	0.0000542
Fat3	3984.4305	-0.3082248	0.0000001	0.0000542
Pbdc1	460.8969	-0.3241598	0.0000002	0.0001879
Heatr6	844.2980	-0.2827739	0.0000003	0.0002263
Fam135b	4116.5733	-0.2007143	0.0000005	0.0003423
Kcnt1	1636.8044	-0.4399229	0.0000012	0.0007474
Jpx	329.7941	-0.4731784	0.0000017	0.0010099
Mtx3	2514.7435	-0.1992080	0.0000020	0.0011185
Cry1	432.7995	0.4671789	0.0000023	0.0012412
Tsc22d3	1915.2891	0.3350705	0.0000025	0.0012839
Zbtb37	1046.5955	-0.2811478	0.0000066	0.0031864
Son	16791.5365	-0.2901293	0.0000115	0.0051209
Zbed6	1849.5321	-0.3919603	0.0000114	0.0051209
Ago3	2865.1364	-0.2359871	0.0000136	0.0053461
Arpc3	3325.7174	0.1704015	0.0000129	0.0053461
Far1	3994.4799	-0.2094132	0.0000134	0.0053461
Rpl3	14232.4971	0.1267615	0.0000160	0.0060513
Rps23	4523.9491	0.1612711	0.0000180	0.0064598
Atxn713b	6748.2832	0.1276009	0.0000183	0.0064598
Kctd13	3795.2485	0.1342014	0.0000211	0.0069393
Vezt	2727.3428	-0.1695772	0.0000206	0.0069393
Eef1a1	63246.1822	0.1271333	0.0000237	0.0075747
Arpc2	9822.3601	0.1357121	0.0000311	0.0082856
Slc25a5	6124.3877	0.1164885	0.0000289	0.0082856
Drap1	3888.9092	0.1479283	0.0000321	0.0082856
Slc20a1	3520.1491	-0.2289946	0.0000325	0.0082856
Hook1	3239.4404	-0.1863168	0.0000283	0.0082856
Klf9	10126.9653	0.1589110	0.0000285	0.0082856
Dzip1	2801.3357	-0.3110676	0.0000309	0.0082856
Mdn1	3006.2392	-0.2859080	0.0000303	0.0082856
Pfcp	8030.7778	-0.1567824	0.0000339	0.0084394
Gm33989	1165.0630	-0.2489200	0.0000361	0.0087705
Ppia	16237.7429	0.1576193	0.0000384	0.0091207
Kmt2c	5357.5891	-0.1918673	0.0000488	0.0106885
Osbpl6	3755.7832	-0.1734559	0.0000492	0.0106885
Per2	1519.9527	0.2236716	0.0000466	0.0106885
Zc3h11a	398.9272	-0.4212390	0.0000484	0.0106885
dync1h1	28032.9115	-0.1932201	0.0000508	0.0108101

With regards to batch effects, we extracted RNA from all the F1 hippocampus samples over multiple rounds and found that “extraction round” was a source of technical variation with batch effect surrogate variable analysis. We therefore controlled for this batch effect in our analysis. The new analysis generated by these updates can be found in Figure 5. The methods and results sections have also been updated accordingly.

Sperm small RNA seq: only 22 small RNAs are affected - this is a very small number across all small RNA classes. I mostly wonder about the piRNA analysis – in my experience it is not meaningful to analyze single piRNAs – yet they should show alterations at the cluster level. Have the authors analyzed the piRNAs as clusters? I cannot imagine a modus operandi for single piRNAs and if such is eluded to it should be substantiated. Also the small RNA sequencing usually doesn't assess the size typical of piRNAs during the library preparation size selection step. The miRNAs could be easily validated by q-PCR?

RESPONSE: In the library preparation stage, the NEXTFLEX small RNA sequencing kit v.4 (shown below) was used. This kit is designed to capture all forms of small RNA from 17 to 65 nucleotides, which includes piRNAs.

As the Reviewer points out, piRNAs are generated from strand specific genomic clusters and therefore it is more biologically meaningful to analyse piRNAs as clusters. Based on this feedback, we have employed a piRNA clustering analysis using the proTRAC piRNA clustering prediction tool (Rosenkranz et al. 2012, *BMC Bioinformatics*, PMID: 22233380). This tool is useful for the prediction, visualization and analysis of piRNA clusters. ProTRAC uses a sliding window approach to detect loci that show high sequence read coverage. Subsequently, sequences mapped to these loci are analyzed with respect to typical piRNA and piRNA cluster characteristics to ensure high specificity. We have applied this approach and have found 4 clusters of piRNAs to be significantly changed. These map onto the gene “Clu” (clusterin) on chromosome 14. The new analysis has been updated and presented in Figure 2. All methods and results sections have also been updated accordingly.

FAQ

Q. What types of RNAs are included in the Small RNA library?

The NEXTFLEX Small RNA Sequencing kit v4 is designed to capture short RNAs (17-65 nucleotides) with 5' monophosphorylated and 3' hydroxylated ends. The kit also allows for the capture of larger molecules (<200 bp) by using the no size selection protocol. MicroRNAs typically have a 5' monophosphate group after cleavage by the enzyme Dicer. Other molecules, such as tRNA and yRNA, also have a 5' monophosphate at their end and they will be incorporated in the libraries if present in the sample, such as serum and plasma. The kit includes blocking oligos to prevent unwanted tRNAs and yRNA from being incorporated into the library. For removal of other types of RNA, please contact our technical support team.

With regards to qPCR validation of RNA-sequencing results, while we are aware of the view that qPCR represents the gold standard in validating transcriptomics results, we argue that this is not necessary in this case, since we used state-of-the-art Illumina next generation sequencing and applied stringent statistical approaches. This means that our RNA-seq data is considered highly reliable. In fact, a recent study demonstrated that there is good concordance between RNA-sequencing results generated by Illumina next generation sequencing and qPCR validation results (Coenye 2021, *Biofilm*, PMID: 33665610). The authors make the point in this editorial that these next generation sequencing approaches should be robust enough as to no longer require qPCR validation provided that there are multiple biological replicates used and stringent analytical approaches such as strict false discovery rate corrections have been applied. We have used sample sizes beyond the minimum numbers seen in many publications, and are therefore confident that we are well powered. Since we have applied all these rigorous methodologies, we believe that qPCR validation is not needed to confirm findings from our robust sequencing analyses. Strongly supporting this view, recent high quality publications the field, including Tomar et al. 2024, did not validate their RNA-Seq results by qPCR (Tomar et al. 2024, *Nature*, PMID 38839949).

Study design: Is it possible to rule out routes of transmission via mating behavior, mother/father carry over of microbiota etc, other immunological remnants of infection (antibodies, Cort levels in the seminal fluid à see Teperino lab studies, or studies by Adam Watson's lab)

RESPONSE: In response to this, we assessed the presence of IgG antibodies to the SARS-CoV-2 spike protein in the male and female breeders using a SARS-CoV-2 spike protein serological IgG ELISA kit in a new cohort. After the breeding period, plasma was taken from female mice bred with either P21 4 weeks post-infection male breeders or mock 4 weeks post-infection male breeders. We also took plasma from the male breeders. When we performed this ELISA (qualitative assessment), we found that all male breeders previously exposed to the P21 SARS-CoV-2 virus had robust expression of IgG antibodies against the SARS-CoV-2 spike protein (absorbance values between 2.1 and 2.8), confirming prior infection with SARS-CoV-2. Importantly, all female breeders that had been exposed to the P21 male breeders had virtually no expression of these IgG antibodies (absorbance values correspond closely to the negative control sample 79), validating that there has been no transfer of infection. Additionally, all mock male breeders and their associated female breeders also had no significant expression of IgG antibodies against the spike protein. This data has been included in a supplementary file and stated in the methods section of the revised manuscript.

Table of blank corrected absorbance values measured at 650nm to determine presence of IgG antibodies against the SARS-CoV-2 spike protein. Blank corrected positive control absorbance = 1.96, and negative control absorbance = 0.097.

Blank corrected absorbance values at 650nm

Male breeders previously mock infected	Blank corrected absorbance values	Male breeders previously infected with P21	Blank corrected absorbance values
80	-0.001	102	2.7945
81	-0.005	103	2.516
75	0	106	2.767
76	0.0105	101	2.6215
77	0.0175	89	2.5325
78	0.0035	104	2.269
94	0.0125	93	2.135
95	0.0025	86	2.618
96	0.0065	88	2.6535
79	-0.0005		
AVERAGE ABSORBANCE	0.00465	AVERAGE ABSORBANCE	2.545222222
Females bred with mock infected mice	Blank corrected absorbance values	SARS exposed females	Blank corrected absorbance values
346	0.0115	343	0.089
344	0.0655	338	0.016
312	0.002	347	0.1175
342	0.006	339	0.07
335	0.0225	345	0.016
330	0.005	340	0.006
336	0.0195	333	-0.003
331	0.0125	328	0.0015
337	0.001	334	0.0135
332	0.0865	329	-0.0045
AVERAGE ABSORBANCE	0.0232	AVERAGE ABSORBANCE	0.0322

RESPONSE: We also investigated mother/father microbiota transfer by performing 16S rRNA sequencing on the fecal samples collected from the female breeders after mating. We found no differences in α -diversity (Shannon Index) between the fecal microbiota of female breeders exposed to mock control male breeders and SARS-CoV-2 (previously exposed) male breeders (**Extended Data Figure 2F**), suggesting a lack of microbiota transfer differences between male and female breeders.

How many litters were used –has this effect of transmission been replicated? Is each offspring n referring to a litter or to a litter mean or to a single offspring animal (is each animal coming from a different litter?). It is currently unclear from the methods section how different timings required to reach peak levels of viral load were accounted for in the mating scheme? Parental age could also be specified further – it currently states: “at least..... “. Were the animals age matched?

RESPONSE: In the F1 generation, there were 10 litters for the paternal SARS-CoV-2 group and 9 litters for the paternal mock infected control group. In the F2 generation, there were 7 litters for the grand-paternal SARS-CoV-2 group and 6 litters for the grand-paternal mock infected control group. This has been updated in the methods section. Each n is referring to an individual animal, however, we accounted for litters in our statistical modelling by including the “litter” each animal came from as a random factor in the linear mixed models for each behavioral outcome. As stated in the “statistical analyses” methods section, litter has been incorporated as a random factor in natural mating studies since there may be decreased phenotypic variation between mice from the same litter.

With regards to age, parental age has been specified further in the methods section. Animals were age matched within each sex, however, males were 12 weeks at the time of breeding whereas females were 8 weeks old.

Peak viral load for this P21 model is typically reached between day 2-4 (usually day 2) post-infection, as described in Bader et al 2023. We can only measure viral load via the lungs (post-mortem dissection) and therefore this is not possible in our paternal paradigm where we need to keep fathers alive. If we were to take blood to measure cytokines everyday, this would add a lot of psychological stress to our male mice which could be a confounding variable in our model. It can be noted that all male mice that were infected reached peak bodyweight loss at three days post-infection and therefore this model is highly consistent. It is preferable in these paternal studies to have a set time between the exposure (in this case the infection event) and the mating period in order to have consistency in terms of which maturing sperm types are affected by the exposure (Tyebji et al. 2020, Gapp et al. 2014, Short et al. 2016). We also reduce variability in our paradigm by only mating those mice that lose at least 9% of their bodyweight on day 3 post-infection.

Microinjection experiment: To me it seems the outcome doesn't partially replicate (one single readout of one behavioral test is mimicking the natural offspring). At most this indicates that the isolated sperm fractions could be functional and should be stated differently. The small RNA enriched sample shows substantial amounts of intact ribosomal RNA that likely indicate somatic cell contamination. It would be important to see all traces prior enrichment and/or other measures of sperm purity to exclude contamination.

RESPONSE: As suggested, we have toned down our statements about recapitulation so that we are stating that these isolated sperm RNA fractions are functional in this paternal paradigm but not necessarily the only mechanism involved in the F1 anxiety-like behavior.

With regards to the enriched fraction, we checked for 18S/28S peaks in each individual sample prior to enrichment, which are all shown below. Samples contain minimal or very small peaks in the 18S or 28S bandwidths.

Samples 11, 13, 15, and 19 are from the SARS-CoV-2 infected mice whereas samples 12, 14,

16, 18 and 20 are from the mock infected control mice.

Comment on sex and gender: The title does currently not reflect that some aspects of the phenotype are sex specific.

RESPONSE: The title has been changed to “Paternal SARS-CoV-2 infection impacts sperm small noncoding RNAs and increases anxiety in offspring in a sex-dependent manner”.

Currently no source data provided

The data availability statement is not conform with nat. comm guidelines. If I as a Reviewer would want to assess the data I'd first have to make a “reasonable request”.

RESPONSE: This has been altered to “uploading the data” upon acceptance of the manuscript. We do not wish to upload our raw data on a public forum before it has been published since we wish to safeguard the originality of our research.

Reviewer #2

The manuscript by Kleeman et al. investigates the very relevant and timely topic of the effects of pre-conceptual exposure to SARS-CoV-2 and its consequences on anxiety-like behavior in mice. In general, the study is well-designed and organized, and I appreciate the substantial amount of work and planning involved in transgenerational studies. However, I have several concerns about the results and data interpretation, as I believe the general conclusions of the study are bold based on the results presented in this manuscript.

I am wondering why the authors decided to report only testicular weight. It would be useful for the study to include some sperm characteristics of infected and non-infected males or to make histological sections of the testes to show if there are differences in the morphological appearance of cells at different stages of spermatogenesis. Additionally, body weight data of the F1 generation should be presented based on sex, as the offspring's sex can be easily determined by anogenital distance from PND 8.

RESPONSE: The bodyweight data of the F1 generation has been presented based on sex from PND 8 onwards (see **Extended Data Figure 1E to 1H**).

As requested here, we have performed histological examination of the testes taken from a new cohort of infected (and mock infected) mice. We counted spermatocytes, round

spermatids and elongated spermatids from sampled seminiferous tubules and have included this data in **Extended Data Figure 2**. We have also updated the methods and results sections accordingly.

I am also curious why the authors measured the whole brain weight rather than evaluating a volume estimation of the hippocampus. This would be a more appropriate readout, as changes in hippocampal volume may reflect structural plasticity that could be associated with behavioral changes. A better approach might be for the authors to dissect the hippocampus bilaterally and then randomly use hippocampi from one side of the brain for RNA sequencing and the other for histology.

RESPONSE: Only fresh tissues were dissected (not perfused). Whole brain weight measurements are fairly simple to do, even within the limitations of our biosafety level 3 facility, and they give a snapshot of any gross anatomical changes in the brain. We did not collect perfused tissues for in-depth volumetric analysis and immunohistochemistry analyses. Heart perfusions are not currently set-up and approved in our PC3 facility (only lung perfusions) and would require extensive validations and testing to pass the Walter and Eliza Hall Institute's institutional biosafety committee regulations. We also wanted to avoid degradation of RNA and weighing individual regions takes a substantial amount of time. We therefore consider this type of volumetric measurement beyond the scope of the current study. However, these are interesting and highly informative experiments to perform in another future study.

Regarding the RNA-seq data, it is surprising that only a small number of differentially expressed genes were found. This could be due to the bulk analysis performed (I understand that single-cell analyses are expensive), but it might be better if the authors attempted to dissect the ventral and dorsal hippocampus separately before running the bulk analysis, as these regions can have distinct gene expression profiles.

RESPONSE: As pointed out here, there can be functional and genomic differences between the dorsal and ventral sections of the hippocampus (Faneslow et al., 2011, *Neuron*, PMID: 20152109). However, in our analysis, despite only a small number of differentially expressed genes in the F1 females (~20), many of these genes have links to anxiety-like behavior as pointed out in the discussion section. Also, many of the papers cited in the discussion use the whole hippocampus for bulk analysis and find many differences between treatment groups. Therefore, analysing whole hippocampus is still a valid approach that can detect relevant differences for our study.

The most interesting part of the study is actually the sperm analysis, even though some key sperm characteristics like morphology, viability, and motility are missing. This section could be presented even before the hippocampal or behavioral data, as it is a crucial part that confirms sperm is affected by SARS-CoV-2 infection. However, due to individual variation between even genetically identical mice, the downside is the pooling of mouse samples for sequencing.

RESPONSE: The sperm RNA section has been moved before the hippocampus RNA section but the behavioural F1 section contains key data related to offspring anxiety (centerpoint of the study as it is in the title). Pooling was necessary due the low RNA yield from sperm from individual mouse samples and is a widely used approach in the paternal epigenetic inheritance literature (Gapp et al., 2014, Tyebji et al. 2020, Tomar et al. 2024, Short et al. 2016).

We have included testes histology analysis and quantification of different germ cell types in **Extended Data Figure 2**. These show that there are no overt fertility changes in the SARS-CoV-2 infected mice.

Also, the behavioral phenotype after the microinjection experiment is a bit more convincing (but again, there is a question about correction for multiple testing). It would also be interesting to see if, after the microinjection experiment, the situation in the hippocampus would remain the same. If the same genes are expressed (even if it's still a small number of genes), that would truly confirm that the results are a consequence of the treatment.

RESPONSE: As suggested here, we performed RNA-seq on the RNA-microinjected hippocampus for males and females. We found very limited changes (only one gene in the males and no genes in the females) suggesting that there was limited recapitulation here, as updated and shown in the new analysis displayed in **Figure 5**. However, other brains regions may be involved in the anxiety-like behavior and perhaps there may be shared transcriptomic changes between F1 and RNA-microinjected brains in these regions.

Therefore, even though this is a truly novel and important experiment, I would advise the authors to reanalyze the behavioral testing, correct fir multiple testing, or use behavioral score, and rewrite the bold statement about the link between infection and anxiety-like phenotypes. Again, as a strong supporter of publishing null findings, since they are equally important, it would be good for the authors to reanalyze the results, rather than focus on finding a couple of genes and making bold statements.

RESPONSE: As suggested, we have re-analysed our behavior with the z-scoring system mentioned above, which is commonly used throughout the literature. This analysis showed a strong link between paternal SARS-CoV-2 infection and offspring anxiety ($F_{(1, 71)} = 9.166$, $P = 0.003$, **Figure 1F**) and therefore we feel our statements about F1 are not too bold. Nonetheless, statements about the sperm RNA-microinjected cohort have been toned down as we saw little recapitulation of the F1 phenotypes in this cohort.

Reviewer #3

It is an interesting paper, and it will have a potential impact on the field. However, some improvements need to be made.

Is hard to interpret Figures 1, 3, and 5. For example, in Figure 1, considering that the only significant values indicated by (*) on the top graph are the values for the male group in Figure 1D and Figure 1L. The post hoc analysis p-values are shown below the graph. I suggest that adding the p-values of significant post hoc analyses to the top of the graph will help to avoid confusion in the interpretation.

RESPONSE: We have added stars and lines for the post-hoc analyses on top of the graphs. The main and interactions effects can be seen below each graph eg. $P_{\text{Paternal Treatment}}$ for main effects. This has been made clear in the figure legends also.

Figure 1L presented in the main figure is not necessary here, it is better to move to the Supplementary figure.

RESPONSE: The graphs have been corrected accordingly: food intake graphs and % bodyweight change graphs have been moved to extended data figures for F1 and F2 generations.

Figure 1O is confusing, the authors say that there is no difference in immune response but on the figure, there are significant changes between control and treatment, so it is better to compare the increased ratios Poly I:C compared to saline in each group.

RESPONSE: All groups show significant acute response to the Poly I:C administration but no significant effect of paternal SARS-CoV-2. We have amended the description to clarify that the acute Poly I:C challenge is working but no effect of paternal treatment on the immune response was observed. We are unable to compare ratios because Poly I:C treated animals are different from saline treated animals.

Surprisingly, the authors determined very few genes that were differentially expressed in F1 females, and no differentially expressed genes were found in males. Based on Figures 1A and 1L, males showed more alterations in behavior. How could the authors explain that?

RESPONSE: We have updated our RNA-sequencing analysis to paired end sequencing and improved our mapping and annotation rates as mentioned earlier in this reply. When we did this, we saw some differences in the F1 male hippocampus of SARS-CoV-2 infected fathers relative to mock infected controls. However, there are less changes in the F1 males when compared to the F1 females. It may be that there are changes in other brain regions not investigated here but this is beyond the scope of this study.

I am also not convinced by the mating time after infection. The full period of spermatogenesis in mice is 7 weeks. Why do the authors mate the mice after 4 weeks after the infection?

RESPONSE: 7 weeks roughly corresponds to the human spermatogenesis cycle but ~35 days corresponds to the mouse spermatogenesis cycle (Griswold 2016, Yoshida 2020, Ernst et al. 2019), hence we mated between 4-5 weeks post-infection (mating started after 4 weeks and went for 6 days).

How was the hippocampus isolated from the whole brain, from fresh or frozen tissue? The precision of dissection could affect the RNA-seq analysis and impact the determination of differentially expressed genes.

RESPONSE: We are highly experienced in the dissection of mouse hippocampus, having dissected these tissues in many previous published studies (Kleeman et al., 2024, Masson et al. 2024, Short et al. 2016, Kuznetsova et al. 2025, Mees et al. 2022). The hippocampus is a structure that is very well delimited, after separating the anterior and posterior connections the hippocampus separates itself easily, after which it can be inspected for contamination from surrounding tissue or connecting tissue. The transcriptomic profiles generated from our F1 hippocampal tissue samples were entirely consistent with the known transcriptomics of the mouse hippocampus, including our own recent work published in this journal (Kleeman et al., 2024, Brain Behav. Immun., PMID: 37820975).

Which quantity of the RNA was taken for RNA-seq analysis?

RESPONSE: 500ng of RNA for mRNA sequencing and 300ng of RNA for small RNA sequencing (AGRF Sequencing guide: https://static1.squarespace.com/static/5c6a2bfa11f7845bc7a99405/t/675119668b44c8632b56a62f/1733368174823/SVG24090RNA_ServiceGuide_RNASequencingService.pdf). This has been updated in the manuscript methods section.

Which Fold Change (FC) cut-off was used for RNA-seq results in both RNA-seq

RESPONSE: Log₂ fold change of 1 or absolute fold change of 2 was used across all datasets for RNA-seq to ensure only biologically meaningful results were reported.

Response to Reviewers

We thank the Editor, and Reviewers, for their highly constructive comments to further improve the latest revision of our manuscript. We are delighted to see that Reviewer #2 and Reviewer #3 were completely happy with the previously revised manuscript, and had no further suggested revision.

In the attached further revised manuscript we have fully addressed each and every one of the latest comments from Reviewer #1 following the previous round of revision of our manuscript. Below we describe how we have described in detail how we have fully addressed each of the latest Reviewer comments.

Reviewer #1

In the revised manuscript titled “Paternal SARS-CoV-2 infection impacts sperm small noncoding RNAs and increases anxiety in offspring in a sex-dependent manner,” the authors Kleeman et al. have made an effort to address all the issues raised. While I find the revised manuscript improved, there are some remaining concerns about the robustness that should be addressed so that I could recommend publication without reservation.

The authors have added important additional clarifications in the text. They have further analyzed weight trajectories in the offspring. Behavior was reanalyzed to summarize individual tests by converting measures into z-scores. The z-score confirmed modest alterations in anxiety behavior (very significant effect, $p=0.003$) and a sex-specific z-score summary of depressive-like behavior. This is interesting given the molecular sex-dependence of hippocampal gene expression. Yet caution is warranted in the interpretation thereof, since the individual test of depressive-like behavior did not show those effects. With the z-score, no power is gained; thus, the authors should explain why the results obtained by computing the z-score in this case doesn't reflect a finding by chance, especially since the F2 offspring shows a similar outcome.

RESPONSE: Integrative z-scoring can reduce behavioural noise which comes from conducting multiple tests across several domains (e.g. anxiety-like, depression-like, etc.) (Guilloux et al. 2012, *J Neurosci Methods*, PMID: 21277897, Kraeuter 2022, *J Neurosci Methods*, PMID: 36435327). It can provide a snapshot of how a mouse is behaving across multiple tests that are related to each other. In that sense, it is an integrative measure and can reveal phenotypes that are not necessarily detected in individual tests due to behavioural noise. For example, the behaviours seen in the depression-like tests (sucrose-preference test and novelty-suppressed feeding test) do not quite reach significance on their own but the integrated z-score reveals that a subtle phenotype is present across the two tests in the F1 male mice. This is because those mice showed convergent behaviours across both tests and the z-score has the sensitivity to reveal this. Mice need to behave in a consistent manner across each test in their respective domains in order for a significant phenotype to be detected. If the mice are randomly showing depression-like behaviour in the novelty-suppressed feeding test (taking longer than control mice to feed on the pellet) but this is not the case in the sucrose-preference test, then the z-score will not be significant. However, the depressive z-score in this case was not seen in our new replication F1 cohort in which we

repeated the main F1 battery (n=10-12 per group), so this effect may not represent a particularly strong or robust phenotype. We toned down our conclusions in relation to this accordingly.

On the other hand, the anxiety measures and anxiety z-score were also seen in our replication cohort, showing comprehensively (across multiple independent large cohorts) that this is a highly robust phenotype.

Overall, even though the authors reduced the number of multiple comparisons by computing z-scores, the issue remains that out of the behavioral phenotypes assessed (light-dark box test, open field test, novelty suppressed feeding, sucrose consumption, social interaction, novel-object recognition, y-maze, summarized into: anxiety, cognition, social interaction, depressive-like, and feeding behavior), only one showed a convincingly significant change (the anxiety behavior). So there is a 20.36% chance of getting at least one false positive; in other words, had this experiment been done five times, once it would show the effect by chance. In that sense, the RNA-injection finding on the anxiety behavior can somewhat strengthen the argument of a robust effect, yet it seems to be sex-dependent. So I am not fully convinced that we are looking at a replicable change. It might be more relevant to see whether those effects can replicate if repeated, rather than relying on the provided z-score analysis, to gauge whether those effects do not reflect a false positive. In the rebuttal, the authors mention that the phenotype has been observed in multiple cohorts, yet the data of both cohorts have been pooled. Seeing the replication in a distinct cohort would make a very strong case and be very convincing. To this end, it would be necessary to see the analysis of both cohorts separately. Also, could the absence of other significant measures not be due to different power (if in anxiety we are looking at 16-22 animals and in the other tests 10-12)?

RESPONSE: The Reviewer makes a valid point here in regards to the replication of our results to ensure robustness. Mating in a PC3 facility can be difficult so we bred multiple cohorts and the second cohort was not fully powered on its own, and therefore we pooled these together. However, in consideration of this comment, we decided to run a separate fully powered (n=10-12 animals per group) replication cohort here to see if the F1 anxiety effects are robust and reproducible.

The results of this cohort have been added to the supplementary section (**Extended Data Figure 3**). Strikingly, we replicated the light/dark box findings in this new cohort, including % light time and latency to enter the light zone. Additionally, we saw a significant difference in light zone entries but we did not see changes in time spent in the centre zone of open field. Importantly, we replicated the F1 anxiety z-score result (P=0.01 main effect of paternal treatment). This result convincingly shows that anxiety is increased in the F1 offspring of SARS-CoV-2 infected males across multiple cohorts.

We also see that all other findings, such as the novel object recognition index, latency to feed in the novelty-suppressed feeding test, and sucrose preference results are not statistically significant in this cohort, whilst the anxiety results are still significant even with the n numbers of 10-12 per group, suggesting that statistical power is not an issue here.

Extended Data Figure 3. Paternal SARS-CoV-2 infection significantly changes anxiety-like behavior in a separate cohort of F1 offspring. Paternal SARS-CoV-2 infection significantly decreases A) % time spent in the light zone of the light-dark box (n=10-12) and B) number of entries into the light zone (n=10-12) in a separate cohort of F1 offspring. Paternal SARS-CoV-2 infection significantly increases C) the latency to enter the light zone in the F1 offspring (n=10-12) without changing D) % time spent in the centre of the open-field (n=10-12) while increasing the E) overall anxiety behavioral z-scores (n=10-12) for F1 offspring. Paternal SARS-CoV-2 has no significant effects on F) the total distance travelled in the open-field (n=10-12), G) the recognition index in trial 2 of the novel-object recognition test (n=10-12), H) % preference for sucrose in the sucrose preference test (n=10-12), and I)

latency to feed in the novelty-suppressed feeding test for F1 offspring (n=10-12). Paternal SARS-CoV-2 does not significantly alter J) overall depression behavioral z-scores (n=10-12), nor does it change K) food consumed over 8-hours after an 18-hour fasting period (n=10-12). Paternal SARS-CoV-2 significantly alters L) % bodyweight changed after a 24-hour fasting period (n=10-12) but does not significantly change M) food consumed over 5 minutes after a 24-hour fasting period (n=10-12). Data presented as mean \pm SEM. General linear models and linear mixed models were used with *post-hoc* analyses where appropriate (Bonferroni-Holm corrected) except for I. Cox regression with proportional hazards was used to analyse I. Each n number refers to the number of individual animals per group. $P_{\text{Paternal Treatment}}$ = main effect of paternal treatment.

How do the authors explain the change in fasting-induced weight loss? Did they check energy expenditure or glucose/insulin tolerance? I do not consider this very crucial - but would be interested to know.

RESPONSE: We did not look at energy expenditure or glucose/insulin tolerance, as metabolism was not the focus of our study. However, we suspect that this result may indicate that the F1 mice have alterations in energy expenditure (e.g. reduced basal metabolic rate or reduced brown fat thermogenesis) or even differences in hormonal regulation after a fast (e.g. leptin, insulin, ghrelin). We think it is unlikely that these mice have reduced activity levels since their locomotion behaviour was not significantly changed in the open field when compared to F1 control mice. However, this does not rule out home-cage activity being reduced, which is also possible. Future studies could investigate this question in more detail, but the focus of our manuscript is on the brain and behavioural phenotypes of the F1 offspring.

The authors made a great addition to their sperm RNA analysis by analyzing their piRNAs at the cluster level. They furthermore predicted putative targets of the single tRNA fragment and miRNA found to be differentially abundant in mature sperm. Can they replicate the miRNA finding by independent sequencing of sperm or by q-PCR in an independent cohort?

RESPONSE: Our sequencing has used sperm from mice sampled from three independent cohorts (this has been added to the Methods section), which increases our confidence that this is a real and robust result. Each replicate contains one mouse from each cohort (n=3 mice total per replicate) so that cohort representation is balanced. With regards to qPCR validation, a recent study demonstrated that there is good concordance between RNA-sequencing results generated by Illumina next generation sequencing and qPCR validation results (Coenye 2021, *Biofilm*, PMID: 33665610). The authors make the point in this editorial that these next generation sequencing approaches should be robust enough as to no longer require qPCR validation provided that there are multiple biological replicates used and stringent analytical approaches such as strict false discovery rate corrections have been applied. We have used sample sizes beyond the minimum numbers seen in many publications, and are therefore confident that we are well powered. Since we have applied all of these rigorous methodologies, we believe that qPCR validation is not needed to confirm findings from our robust sequencing analyses. Strongly supporting this view, recent high-quality publications the field, including Tomar et al. 2024, did not validate their RNA-Seq results by qPCR

(Tomar et al. 2024, *Nature*, PMID 38839949). If we were to go ahead with qPCR, it would make more sense to validate all of the piRNAs, tRNA and miRNA, which would be very costly and difficult to do. Given that this result has come from three independent cohorts, we believe that it is robust.

The authors relate the behavioral and molecular sex-dependent findings. Support of this conclusion would require seeing that some of the gene expression changes are indeed involved in the anxiogenic phenotype following the same pattern.

RESPONSE: Since the molecular sex-dependent findings only correlate with the behaviour, we have revised our conclusions text surrounding the relationship between our molecular and behavioural findings. In particular, we have deleted the sentence “It is therefore possible that the reduced expression of hippocampal genes identified here in the F1 female offspring, most notably *Prl*, may be at least in part associated with the increased anxiety observed in these mice”, as we do not have causative evidence for this.

It is great that the authors conducted another round of gene expression analysis – how did it compare to the first round? Even if mapping rates were worse before, were some of the prior findings replicable? Thanks for having clarified what was referred to by batch effects.

RESPONSE: In the F1 females, we do see some overlap in differentially expressed hippocampal genes between the original gene expression analysis and the revised gene expression analysis. Overall, the two analysis rounds have six differentially expressed genes in common showing that some of the prior findings were replicable. Interestingly, in the first analysis, the prolactin receptor was found to be downregulated in the females but in the second round, the hormone itself (prolactin) was found to be downregulated. However, the rest of the differentially expressed genes do not overlap (14/20 for first analysis, 9/15 for second analysis).

Here is a table comparing the differentially expressed genes that were common between the two rounds of analysis:

Round 1	Round 2
Calpain 11 (Capn11 , upregulated, P=0.02)	Calpain 11 (Capn11 , upregulated, P=0.000034)
Orthodenticle homeobox 2 (Otx2 , downregulated, P=0.01)	Orthodenticle homeobox 2 (Otx2 , downregulated, P=0.000034)
Collagen type VIII alpha 2 chain (Col8a2 , downregulated, P=0.01)	Collagen type VIII alpha 2 chain (Col8a2 , downregulated, P=0.000034)
Aquaporin 1 (Aqp1 , downregulated, P=0.01)	Aquaporin 1 (Aqp1 , downregulated, P=0.00047)
WD repeat domain 86 (Wdr86 , downregulated, P=0.04)	WD repeat domain 86 (Wdr86 , downregulated, P=0.0049)
Coagulation factor V (F5 , downregulated, P=0.002)	Coagulation factor V (F5 , downregulated, P=0.029)

It is interesting and fitting that some differentially regulated genes are known to play a role in anxiety-related behaviors. If there is a sex-independent behavioral effect at the z-score level related to hippocampal gene expression and those genes are indeed involved in the present model, why would they not be differentially regulated in the males as well? Do they show the same tendency? Are they known to be sex-dependently regulated? If the gene expression is analyzed for an interaction between sex and treatment, no significant genes are revealed – to me that means that no independent analysis separated by sex is indicated. Should the differentially regulated genes be at all related to the observed anxiety behavior, why does the RNA injection not recapitulate more of those changes?

RESPONSE: Many genes found to be differentially expressed in the F1 females are sex-dependently regulated and do not show a tendency towards differential expression in the F1 males. Genes related to the prolactin system are known to be sex-dependently regulated. In fact, *Prl* (prolactin gene) was filtered out in the male F1 analysis during the filtering stage due to extremely low expression. While prolactin can be present in males, it is typically found at much lower levels so we do not expect it to be differentially expressed in the F1 males. Many of the other genes found to be differentially regulated in the F1 females, including *Capn11*, *Gh*, *Slc13a4*, *Aqp1*, *Otx2*, *F5*, *Pcolce*, *Steap1*, have also been shown to be sex-dependently regulated in the mouse hippocampus (Tchessalova et al. 2020, *Neuroscience*, PMID: 31917350). As pointed out before, it is a valid approach to separate the sexes in an RNA-seq experiment and this is commonly seen in the literature (Fiorini et al. 2022, *Front Neurol*, PMID: 36619915, Rai et al. 2022, *J Orthop Res*, PMID: 35266580), especially in the hippocampus where there is sexual dimorphism.

It is likely that other molecular and cellular changes in other brain regions are associated with the anxiety-like behaviours present in these F1 mice. We have toned down the speculative commentary about these genes being linked to the anxiety phenotype since we do not see shared hippocampus transcriptomic changes between the sexes. However, these results do suggest that there are subtle sex-dependent changes occurring in the offspring hippocampal gene expression as a result of paternal SARS-CoV-2 infection. **This in and of itself is interesting since it suggests that paternal SARS-CoV-2 exposure may be influencing gene expression patterns from early in development.**

In terms of the rebuttal response to the point of why extreme cases should or should not be mentioned in the introduction, the authors argue that the model does produce those cases. That is not the point of my criticism. The point is: is the present study using those severe cases? If not, why are they brought into the picture? Even if the mice used in the study are simply too young to see those severe phenotypes, then those severe phenotypes are not the subject of the current study and should not be emphasized. But this is not an essential point to me - I simply find it misleading.

RESPONSE: Since we chose mice that lost at least 9% of their bodyweight at the peak of the SARS-CoV-2 infection (but lower than 20% since this is a humane endpoint), they would fall under the category of a moderate infection. In light of this point, we have restructured this section of the introduction so that we do not emphasize severe disease.

For less affected cases as studied in the manuscript: even if the sperm RNA profile is

distinct from the one produced by stress alone – in this model, the sperm RNA profile does not seem to be a strong driver of intergenerational change in the main anxiety z-score, nor are the sperm RNA changes particularly strong (with piRNA changes appearing to show the strongest effects, yet their involvement in mammalian transmission is yet to be proven in the field). Samples 14, 15, and 18 don't look consistent with the other samples in that they show considerable ribosomal RNA peaks – as high as the small RNA fraction. Do the authors have orthogonal measures of purity? What samples do those "outliers" correspond to in the heatmap Fig.2. The pasted bioanalyzer plots were too blurry to read this accurately.

RESPONSE: Sample 14 is control_2, sample 15 is SARS_3, and sample 18 is control_4. We also used RIN (RNA integrity) numbers as a guide for somatic cell contamination and RNA degradation. Below are the RIN numbers for our samples. As a guide, RIN numbers below 5 (and >2) are generally considered to be ideal for sperm RNA (Tiwari et al. 2023, *Biochem Cell Biol*, PMID: 37948675). RIN numbers above this are considered to have somatic cell contamination:

11 (SARS_1): 4.3
12 (control_1): 4.0
13 (SARS_2): 4.9
14 (control_2): 5.2
15 (SARS_3): 4.6
16 (control_3): 3.7
18 (control_4): 4.1
19 (SARS_4): 3.5
20 (control_5): 3.5

Sample 14 is on the borderline but we considered it important to include in order to be well-powered.

We have also enlarged the bioanalyser plots below.

I appreciate the authors recognizing the limitation with regards to the distinction between body weight-induced effects and infection-induced effects. I wonder though whether comparing the sperm RNA levels here is meaningful again due to their apparently limited involvement in the transmission even if functional. At the same time, to my understanding, the authors did not assess metabolic alterations in the current model in the offspring which could be present, and as the authors pointed out, their own prior model also observed behavioral effects in the offspring, even though a comparison of models seems hard, since maternal behavior was affected in the nutritional model. Currently, to me, it remains hard to conclude from the present data that the infection here is different from body weight-induced changes. Pointing this out as a limitation in writing is important, yet I am not convinced that is sufficient. Can the authors try to adjust offspring behavioral measures with the paternal weight as a covariate? Showing a persistent effect would make a strong case.

RESPONSE: In response to this, we ran linear regression/correlation analyses on our robustly significant behavioural measures (seen across multiple cohorts) to assess whether paternal peak bodyweight loss correlated with any of the measures. We found no significant correlations between paternal bodyweight and any of the main measures. The results are displayed below in Response to Reviewer Figure 1. We believe that this indicates that the severity of paternal weight loss does not significantly influence our main significant outcome measures related to F1.

Response to Reviewer Figure 1. Linear correlations between paternal bodyweight loss (at the peak of infection) and significant outcomes measures from F1.